# Developmental mouse brain common coordinate framework

Fae N. Kronman[1], Josephine K. Liwang[1], Rebecca Betty[1], Daniel J. Vanselow[2], Yuan-Ting Wu[1], Nicholas J. Tustison[3], Ashwin Bhandiwad[4], Steffy B. Manjila[1], Jennifer A. Minteer[1], Donghui Shin[1], Choong Heon Lee[5], Rohan Patil[1], Jeffrey T. Duda[6], Jian Xue[7], Yingxi Lin[7], Keith C. Cheng[2], Luis Puelles[8], James C. Gee[6], Jiangyang Zhang[5], Lydia Ng[4] & Yongsoo Kim[1] ✉

3D brain atlases are key resources to understand the brain's spatial organization and promote interoperability across different studies. However, unlike the adult mouse brain, the lack of developing mouse brain 3D reference atlases hinders advancements in understanding brain development. Here, we present a 3D developmental common coordinate framework (DevCCF) spanning embryonic day (E)11.5, E13.5, E15.5, E18.5, and postnatal day (P)4, P14, and P56, featuring undistorted morphologically averaged atlas templates created from magnetic resonance imaging and co-registered high-resolution light sheet fluorescence microscopy templates. The DevCCF with 3D anatomical segmentations can be downloaded or explored via an interactive 3D web-visualizer. As a use case, we utilize the DevCCF to unveil GABAergic neuron emergence in embryonic brains. Moreover, we map the Allen CCFv3 and spatial transcriptome cell-type data to our stereotaxic P56 atlas. In summary, the DevCCF is an openly accessible resource for multi-study data integration to advance our understanding of brain development.

Brain atlases provide a standard anatomical context to interpret brain structure[1,2] and function, including neuronal connectivity[3–6], molecular signatures[7–9], and cell type specific transcriptome data[10,11]. Atlases contain two related yet independent components: templates with distinct contrast features and anatomical delineations. Historically, mouse brain atlases existed in the form of 2-dimensional (2D) histologically stained sections and annotations based on cytoarchitecture often from a single male specimen[12,13]. These atlases present challenges in interpreting anatomical regions in 3-dimensional (3D) brain imaging data and may misrepresent anatomy across different individuals and sexes[14–16]. Moreover, recent advancements in cellular resolution whole mouse brain imaging techniques allow researchers to rapidly produce increasing amounts of 3D data from mouse models at various ages[6,17–19], calling for 3D reference atlases for data analysis[20,21]. Currently, the Allen Institute adult mouse brain common coordinate framework (CCFv3) serves as a 3D reference atlas to overcome many limitations of 2D atlases described above and to provide standardized spatial context to integrate data from different studies for the adult mouse brain[15]. Recent brain mapping tools, such as mBrainAligner[22,23] and Multimodal 3D Mouse Brain Atlas Framework with the Skull-Derived Coordinate System[24], have enabled cross modal mapping of MRI, LSFM, and other 3D datasets to the CCFv3, advancing the use of this mouse atlas[6,25].

[1]Department of Neural and Behavioral Sciences, College of Medicine, The Pennsylvania State University, Hershey, PA, USA. [2]Department of Pathology, College of Medicine, The Pennsylvania State University, Hershey, PA, USA. [3]Department of Radiology and Medical Imaging, University of Virginia, Charlottesville, VA, USA. [4]Allen Institute for Brain Science, Seattle, WA, USA. [5]Bernard and Irene Schwartz Center for Biomedical Imaging, Department of Radiology, New York University School of Medicine, New York, NY, USA. [6]Department of Radiology, Penn Image Computing and Science Lab, University of Pennsylvania, Philadelphia, Pennsylvania, USA. [7]Department of Psychiatry, University of Texas Southwestern Medical Center, Dallas, TX 75390, USA. [8]Department of Human Anatomy and Psychobiology, Faculty of Medicine, Universidad de Murcia, and Murcia Arrixaca Institute for Biomedical Research (IMIB), Murcia, Spain. ✉e-mail: yuk17@psu.edu

However, we lack high resolution 3D mouse brain atlases that account for developmental anatomy as well as ontologically consistent segmentations that can be used from fetal to adult brains. Developing brains undergo rapid shape and volume changes with cell proliferation and migration, guided by regionally distinct gene expression[26–31]. Therefore, developmental research requires a standard reference atlas to comprise several templates and annotations that span multiple stages of development and remain connected by a common anatomical schema. Several developmental atlases have been published varying by imaging modality, accessibility, age ranges, labeled detail, and template specifications (Table 1). The Allen Developing Mouse Brain Atlas (ADMBA) is a commonly used reference, consisting of a series of 2D histological sections with annotations based on the prosomeric model using an uninterrupted series of transverse brain segments[28,32]. Yet, ADMBA is based on a single animal per age and offers limited usage to accommodate emerging high-resolution 3D whole brain data. Although recent 3D atlases interpolated and extrapolated from the ADMBA helped to generate 3D templates and annotations[33], single modality-based templates and smoothed annotations without biological validation require further improvement. Moreover, existing MRI-based developmental mouse brain atlases do not meet community needs for cellular resolution 3D imaging and conflicting annotations across different atlases present major challenges to interpret brain areas across different studies[14,34–37].

Here, we generated a developmental common coordinate framework (DevCCF) for mouse brains with expert-curated 3D segmentations at seven ages: embryonic day (E)11.5, E13.5, E15.5, E18.5, postnatal day (P)4, P14, and P56. Each age features undistorted morphology and intensity averaged symmetric templates from both male and female brains with at least four distinct magnetic resonance imaging (MRI) contrasts, as well as a light sheet fluorescence microscopy (LSFM) based template. Furthermore, we established developmentally consistent annotations based on the prosomeric model of mammalian brain anatomy[32]. We also integrated the CCFv3 and P56 DevCCF templates, which enables users to compare the two distinct and complementary anatomical labels in the same space. We demonstrated the utility of the DevCCF by mapping GABAergic neurons and spatial transcriptome cell type data from developing and adult mouse brains, respectively. Lastly, we established an interactive viewer and downloadable datasets to freely distribute the DevCCF (https://kimlab.io/brain-map/DevCCF/).

## Results
### Pipeline overview to create the DevCCF
Creating each of the seven DevCCF atlases requires three primary steps: (1) 3D symmetric MRI and LSFM template generation, (2) multimodal template alignment (registration), and (3) anatomical segmentation. 3D symmetric templates for each modality are intensity and morphological averages of individual samples generated using non-linear registration with Advanced Normalization Tools (ANTs v2.3.5; Fig. 1a)[38]. We used LSFM to acquire high resolution images (up to 1.8 μm/voxel) with cell type specific labeling and other staining to acquire cellular features. We also used various MRI contrast datasets (e.g., diffusion weighted imaging; DWI) of ex-vivo in-skull mouse brain samples to create morphologically undistorted MRI templates with up to 31.5 μm isotropic voxel resolution (Fig. 1a). To mitigate morphological distortion due to removal of the brain from the skull and

## Table 1 | Features of developmental mouse brain atlases

| Atlas | Citation | Modality | Samples per template | Resolution (μm) | Labels | Age(s) |
|---|---|---|---|---|---|---|
| Chemo-architectonic Atlas of the Developing Mouse Brain[108] | Jacobowitz and Abbott, 1997, CRC Press | 2D Cytoarchitecture | 1 | - | 105 plates | E11-12 E13-14 E15-16 E17-18 P0 |
| Atlas of the Developing Mouse Brain[109] | Paxinos et al., 2007, Academic Press | 2D Cytoarchitecture | 1 | - | 43 plates 65 plates 73 plates | E17.5 P0 P6 |
| Prenatal Mouse Brain Atlas[110] | Schambra et al., 2008, Springer New York | 2D Cytoarchitecture | 1 | 10 μm slice interval | 56 plates 57 plates 63 plates 58 plates | E12 E14 E16 E18 |
| MRI and μCT Combined Atlas of Developing and Adult Mouse brains[78] | Aggarwal et al., 2009, Methods in Molecular Biology | 3D MRI, μCT | 1 MRI \| 1 μCT 1 MRI \| 1 μCT 1 MRI \| 1 μCT 1 MRI \| 1 μCT 1 MRI \| 1 μCT 9 MRI \| 8 μCT | 125 μm isotropic | 35 (P140+ only) | P7 P14 P21 P28 P63 P140+ |
| MRI Atlas and Database of the Developing Mouse Brain[34] | Chuang et al., 2011, NeuroImage | 3D MRI | 5 4 7 10 | 80 μm isotropic 90 μm isotropic 100 μm isotropic 125 μm isotropic | 16 24 24 35 | E16 P0 P7 P80+ |
| Allen Developing Mouse Brain Atlas[28] | Thompson et al., 2014, Neuron | 2D Genoarchitecture | 1 | 16 ×16 x 20 μm³ 16 ×16 x 20 μm³ 16 ×16 x 20 μm³ 16 ×16 x 20 μm³ 20 ×20 x 20 μm³ 20 ×20 x 25 μm³ 20 ×20 x 25 μm³ 20 ×20 x 20 μm³ | 239 703 729 73 76 79 928 921 | E11.5 E13.5 E15.5 E18.5 P4 P14 P28 P56 |
| MEMRI atlas of neonatal FVB/N mouse brain development[37] | Szulc et al., 2015, NeuroImage | 3D MRI | 6 | 100 μm isotropic | 15 | P1 – P11 (daily) |

Developmental mouse brain atlases can be compared by features including biological information, imaging modality, number of samples per template, template resolution, and number of plates or segmented regions, and ages represented. Acronyms: micro computed tomography (μCT).

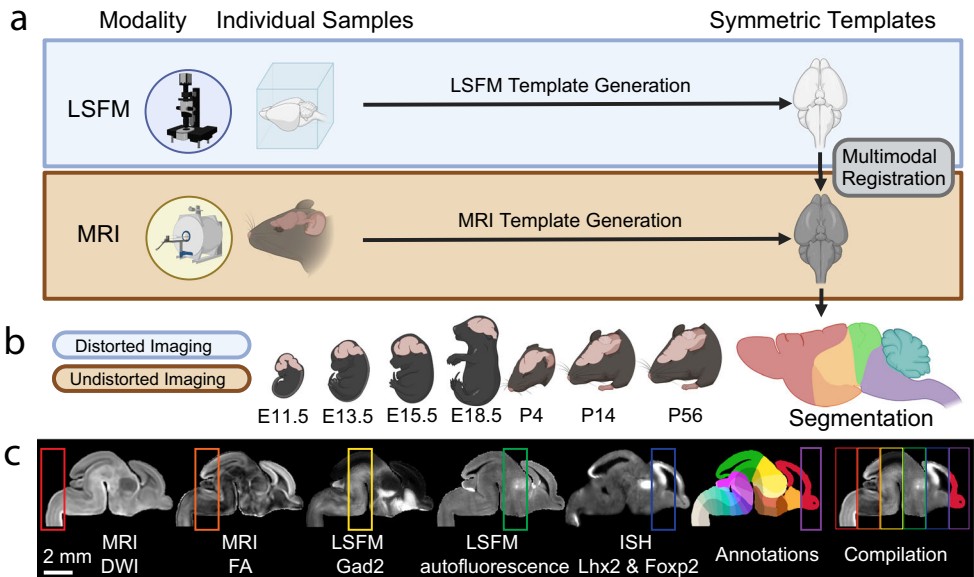

**Fig. 1 | DevCCF overview. a** Morphology and intensity averaged templates from light sheet fluorescence microscopy (LSFM; blue box) and magnetic resonance imaging (MRI; orange block) imaging. The LSFM template was aligned to the undistorted MRI template via multimodal registration. **b** Multimodal DevCCF with anatomical segmentations is established at four embryonic and three postnatal ages. **c** Sagittal slices of multimodal E15.5 data registered to the E15.5 DevCCF, each highlighting unique anatomical features. Data includes MRI DWI (red), MRI FA (orange), LSFM GABAergic neurons from *Gad2*-Cre;Ai14 mice (yellow), LSFM autofluorescence (green), in situ hybridization (ISH) Lhx2 and Foxp2 gene expression (blue), DevCCF annotations (violet), Compiled colocalized data from each data type, as labeled. Scale bar = 2 mm. **a**, **b** created in BioRender. Kim, Y. (2023) BioRender.com/q01q598.

## Table 2 | DevCCF template descriptors by age

| Age | MRI samples (female, unknown) | LSFM samples (female) | MRI resolution[a] (μm isotropic) | Template contents | Mean brain volume ± SD (mm³) | Number of labels |
|-----|-------------------------------|------------------------|----------------------------------|-------------------|------------------------------|------------------|
| E11.5 | 6 (1, 1) | 10 (5) | 31.5 | Embryo | 5.32 ± 0.55 | 77 |
| E13.5 | 9 (3, 1) | 10 (5) | 34 | Head (MRI) Embryo (LSFM) | 19.01 ± 1.00 | 98 |
| E15.5 | 9 (2, 4) | 9 (4) | 37.5 | Head (MRI) Embryo (LSFM) | 41.69 ± 5.26 | 141 |
| E18.5 | 8 (3,0) | 9 (4) | 40 | Head (MRI) Brain (LSFM) | 83.98 ± 3.50 | 146 |
| P04 | 10 (0,9) | 7 (4) | 50 | Brain | 192.15 ± 22.02 | 192 |
| P14 | 14 (5,0) | 10 (5) | 50 | Brain | 380.48 ± 8.54 | 288 |
| P56 | 10 (5,0) | 6 (3) | 50 | Brain | 435.77 ± 11.49 | 288 |

Template descriptors include sample size used to develop each template by sample sex, resolution of MRI templates, template contents (whole embryo, whole head, or whole brain), mean brain volume and standard deviation, and the number of annotated labels at each age. Mean brain volume calculated from source data file for Supplementary Fig. 5. [a]DevCCF template resolution is given for MRI templates. Aligned LSFM templates and annotations are 20 μm isotropic.

subsequent sample processing (e.g., tissue clearing), LSFM templates were non-linearly registered to the MRI templates, generating multi-modal DevCCF templates with cellular resolution features and undistorted morphology (Fig. 1a). Moreover, we established developmentally consistent anatomical segmentations based on cyto- and geno-architecture at seven developmental ages (E11.5, E13.5, E15.5, E18.5, P4, P14, and P56; Fig. 1b)[28,32,39]. Finally, we employed the DevCCF to map gene expression and other cell type information from various 2D and 3D imaging modalities to guide and validate our anatomical segmentations (Fig. 1c).

### 3D multimodal developmental mouse brain templates
We used MRI to image paraformaldehyde-fixed ex-vivo in-skull samples at 31.5 μm (E11.5), 34 μm (E13.5), 37.5 μm (E15.5), 40 μm (E18.5), and 50 μm isotropic nominal voxel resolution (P4, P14, and P56) from both male and female mice (Table 2). Using MRI DWI contrast for embryonic samples and apparent diffusion coefficient (ADC) contrast for postnatal samples, we iteratively registered age matched sample data and

their midline reflections to their composite average to create morphology and intensity averaged symmetric templates (Sample size and template details in Table 2; Individual sample details in Supplementary Data 1). We applied identical transformation fields and averaging procedures to each sample MRI contrast (transverse relaxation time weighted; T2-weighted, fractional anisotropy; FA, DWI, and ADC) to create a minimum of four symmetric MRI templates with distinct contrasts at each age (Fig. 2a). Moreover, we utilized tissue clearing and LSFM to image either the whole embryo (E11.5, E13.5, and E15.5) or extracted whole brain (E18.5, P4, P14, and P56) at resolution of 1.8 (x) x 1.8 (y) x 5.0 (z) μm³/voxel (Table 2). LSFM data were resampled to 10 μm isotropic voxel resolution to generate symmetrical templates (Fig. 2b) as done for MRI templates. Averaging co-registered samples results in DevCCF templates with reduced individual noise and enhanced regional boundaries (Supplementary Fig. 1). Moreover, symmetrical templates can serve as initial space to assess potential laterality and sexual dimorphism of the brain by registering to individual samples.

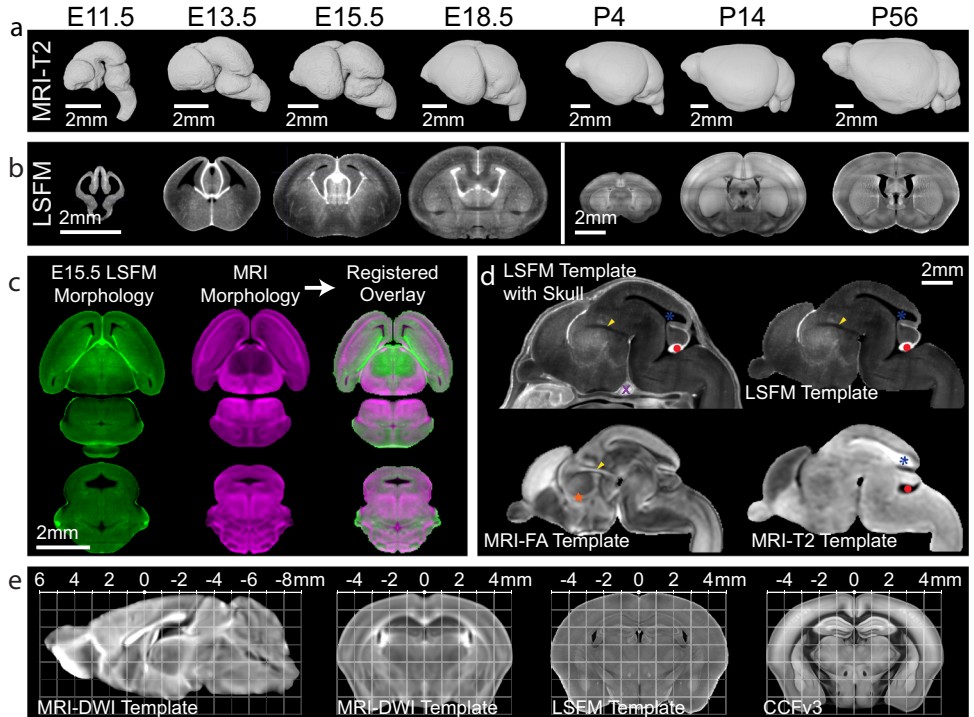

**Fig. 2 | 3D Multimodal developmental mouse brain templates. a** 3D DevCCF MRI templates from T2-weighted contrasts. **b** DevCCF LSFM autofluorescence templates (coronal slice) before multimodal registration to the MRI template. **c** Multimodal registration of E15.5 LSFM (green) to MRI DWI (magenta) templates. Top: horizontal plane. Bottom: coronal plane. **d** DevCCF E15.5 multimodal templates with unique and complementary contrasts. Note distinct anatomical marks differentially highlighted: purple 'X' (trigeminal nerve root), red circle (choroidal tissue), orange star (zona limitans), yellow arrowhead (retroflex tract), blue asterisk (ventricle). **e** The CCFv3 template is registered to the P56 DevCCF. Images are MRI DWI template midline sagittal slice, co-registered MRI DWI, LSFM autofluorescence, and CCFv3 coronal template coronal slices, as labeled. Scale bars in **a**–**c** = 2 mm.

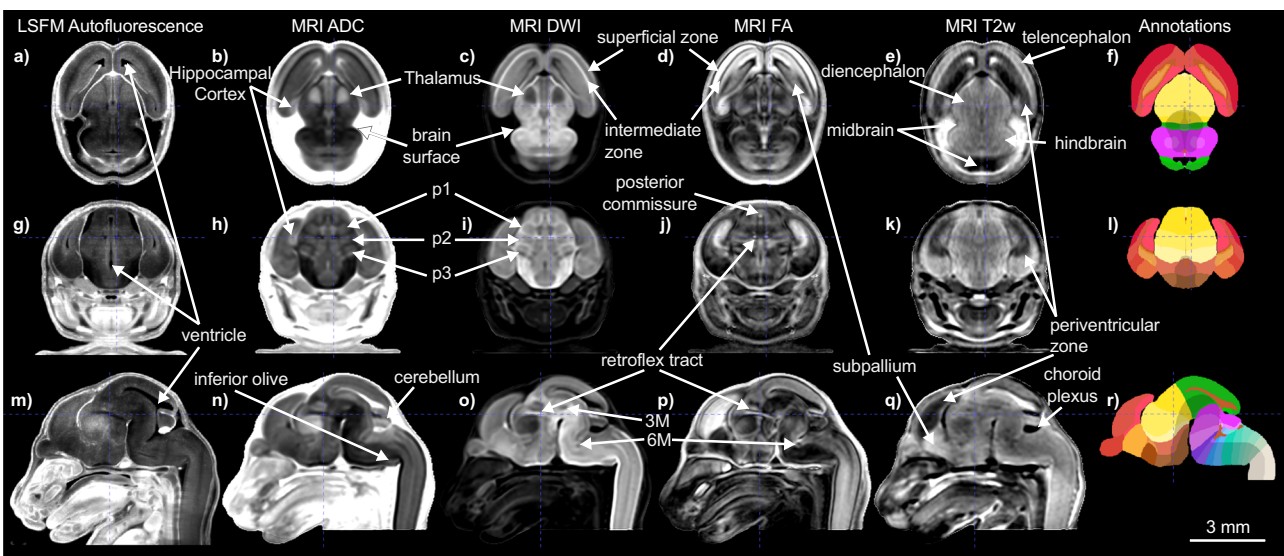

**Fig. 3 | Example of landmarks used for atlas parcellation at E13.5.** The E13.5 DevCCF includes 6 aligned 3D whole brain images: an LSFM Autofluorescence template, 4 MRI contrast templates, and developmental annotations. We show each template as horizontal (**a**–**f**), coronal (**g**–**l**), and sagittal (**m**–**r**) cross sections with a selected set of visible landmarks that guided atlas segmentation. Selected landmarks are noted, including the dark ventricles marked in the LSFM template (**a**, **g**, **m**), telencephalic intermediate and superficial layers with inverse contrasts in DWI and FA templates (**c**, **d**), the telencephalic periventricular zone in the T2w template (**e**, **k**, **q**), the retroflex tract that denotes the boundary between prosomeres (p)2 and 3 in DWI and FA templates (**j**, **o**, **p**), and distinct diencephalic prosomeres 1, 2, and 3 in MRI ADC and DWI templates (**h**, **i**), the cerebellum in all sagittal viewed templates (**m**–**r**). Scale bar = 3 mm. 6M abducens motor nucleus, 3M occulomotor nucleus. Annotation colors are defined in DevCCF ontology (Supplementary Data 2).

## Table 3 | Segmentation landmarks

| Neuromere | Landmarks |
|---|---|
| hypothalamic prosomere 1 (hp2) | mammillary area (Mam)[a]<br>supraoptic commissure (soc)[b]<br>preoptic area (PO)[b]<br>anterior commissure (ac)[b] |
| hypothalamic prosomere 1 (hp1) | retromammillary area (RMa)[a]<br>subthalamic nucleus (STh)[a],<br>fornix tract (fx) |
| prosomere 3 (p3) | rostral ventral tegmental area (p3VTA)[c]<br>rostral substantia nigra (p3SNR, p3SNC)[a] |
| prosomere 2 (p2) | retroflex tract (rf)<br>habenula (Hb)[b]<br>thalamus (p2A)[b]<br>habenular commissure (hbc)[d]<br>pineal gland and stalk (Pi)[d] |
| prosomere 1 (p1) | posterior commissure (pc)[b] |
| mesomere 1 (m1) | ventral tegmental decussation (vtg)[a]<br>dorsal tegmental decussation (dtg)[a]<br>oculomotor nucleus (3 M)[a]<br>inferior colliculus (IC)[b]<br>superior colliculus (SC)[b] |
| mesomere 2 (m2) | Rostral-most r0 to caudal-most m1 landmarks |
| rhombomere 0 (r0) | prodromal interpeduncular nucleus (MnIPPr[c], IPPr[a])<br>caudal ventral tegmental area (r0VTA)[c]<br>caudal substantia nigra (r0SNC, r0SNR)[a]<br>decussation of superior cerebellar peduncle (dscp)[a]<br>trochlear nucleus (4 M)[a]<br>cerebellar vermis (CbV)[b] |
| rhombomere 1 (r1) | interpeduncular nucleus (IPR, IPC)[a]<br>dorsal tegmental nucleus (DTg)[a]<br>cerebellar hemisphere (CbH)[b] |
| rhombomere 2 (r2) | rostral end of trigeminal motor nucleus (r2-5M)[b] |
| rhombomere 3 (r3) | rostral pontine nucleus (r3LPn, r3MPN)[a]<br>caudal end of trigeminal motor nucleus (r3-5M)[b] |
| rhombomere 4 (r4) | caudal pontine nucleus (r4LPn, r4MPN)[a]<br>efferent facial nerve (7ne) |
| rhombomere 5 (r5) | trapezoid body (tz)[a]<br>superior olive (MSO, LSO)[a]<br>abducens motor nucleus (6 M)[a]<br>genu of facial nerve (7 ng) |
| rhombomere 6 (r6) | ascending facial nerve (7na)<br>facial motor nucleus (7 M)[b] |
| rhombomere 7 (r7) | Rostral-most r8 to caudal-most r6 landmarks |
| rhombomere 8 (r8) | rostral inferior olive (r8-IO)[a] |
| rhombomere 9 (r9) | middle inferior olive (r9-IO)[a] |
| rhombomere 10 (r10) | middle inferior olive (r10-IO)[a]<br>middle hypoglossal motor nucleus (r10-12M)[a] |
| rhombomere 11 (r11) | caudal inferior olive (r11-IO)[a] |

Neuromeres are listed from rostral to caudal with frequently used landmarks for segmentation.
[a]Basal plate landmark.
[b]Alar plate landmark.
[c]Floor plate landmark.
[d]Roof plate landmark.

To combine the benefits of undistorted MRI morphology and LSFM cellular resolution, we optimized 3D landmark assisted multi-modal registration methods to map LSFM templates to age matched MRI templates at 20 μm isotropic resolution (Fig. 2c, d; see Methods). Unique imaging contrasts of each template in the same morphology highlight distinct anatomical structures (Figs. 2d and 3). Moreover, we established the postnatal templates with stereotaxic coordinates by

aligning estimated bregma and anterior commissure locations from MRI templates (Supplementary Fig. 2a). 3D stereotaxic coordinate grid images (Supplementary Fig. 2b) were generated for each postnatal age that can be overlaid on templates to facilitate in vivo injections or recording experiments in the future[40,41]. Lastly, we conducted multi-modal registration to establish the widely used CCFv3 template and its annotations in the stereotaxic system of the P56 templates to promote seamless integration of CCFv3 mapped data onto the P56 DevCCF template (Fig. 2e).

### 3D Anatomical labels with an updated developmental ontology

The DevCCF ontology provides an update to the prosomeric ADMBA 13-level hierarchical ontology[28,32] to accommodate advancements in our understanding of developing vertebrate brain anatomy, such as concentric ring topology in the pallium (Supplementary Fig. 3)[39]. It also highlights data-driven modifications to the ADMBA ontology, such as renaming the isthmus to rhombomere 0[42]. Nevertheless, the DevCCF ontology closely follows the structure and information of the prosomeric ADMBA ontology, matching most region names, abbreviations, and color assignments (Supplementary Data 2)[28].

We used landmarks (Table 3) that are visible in DevCCF templates (Fig. 3, Supplementary Fig. 4), as well as aligned 3D imaging and side-by-side reference materials as evidence to draw 3D anatomical segmentations. Several template contrasts help delineate the brain surface-to-surface areas at the choroidal tissue, pallium, and brainstem (Fig. 3a, g, m) and cerebellum and midbrain (Fig. 3m), as well as surface to skull areas (Fig. 3n–r). Further segmentation was continued at increasing ontological depth, to at least Level 5 (Fig. 4a).

The DevCCF segmentations are simplest at E11.5 and grow in complexity and number through P56 (Fig. 4b, c; Supplementary Data 2). E11.5 annotations consist of ventricles, neuromeric boundaries in the rostral to caudal direction; floor, basal, alar, and roof boundaries in the ventral to dorsal direction; and pallium, subpallium divisions of the telencephalon (Fig. 4b, c). E13.5 annotations include additional subpallial segmentations as well as early stages of the concentric ring topology, defining early divisions of the neocortex, allocortex, and mesocortex (Fig. 4b, c). Cortical layers are segmented as early as P4, and cerebellar layers are segmented as early as P14 (Fig. 4b, c).

Segmentation was aided and validated by 3D histological staining from LSFM with alignment to the DevCCF (Fig. 4d–f). For example, neurofilament staining supports neocortical layer delineation, with the strongest staining in layer 4, and distinguishes mesocortical olfactory, insular, hippocampal, and cingulate cortices from each other (Fig. 4e). Aligned fluorescence Nissl images similarly aided P56 the delineation of cerebellar internal granular layer, and olfactory bulb layers delineation (Fig. 4f). Each segmentations are made to be mutually exclusive, meaning all brain voxels are labeled uniquely with a single ontology ID (Supplementary Data 2).

Lastly, we imported 2D sectioned ADMBA in situ hybridization (ISH) gene expression data onto age matched DevCCF templates to validate our anatomical segmentation. We confirmed that gene expression profiles registered to the DevCCF are concordant with our template signal and anatomical segmentations at each developmental age (Supplementary Fig. 4).

The DevCCF with 3D labels offers unique opportunities to quantify the volumetric growth curve of normally developing C57bl/6 mouse brains. We calculated the individual regional volumes and their growth trajectories from E11.5 through P56 (Fig. 4g; Supplementary Fig. 5). The plots show sigmoidal growth curves with rapid volume expansion around birth with delayed cerebellar development (Fig. 4g; Supplementary Fig. 5)[34]. DevCCF whole brain volume calculated from template annotations are within one standard deviation of whole brain volume based on the average individual samples, except at P56, which is 1.08 standard deviations from the mean (black circles in the Fig. 4g, Supplementary Data 3).

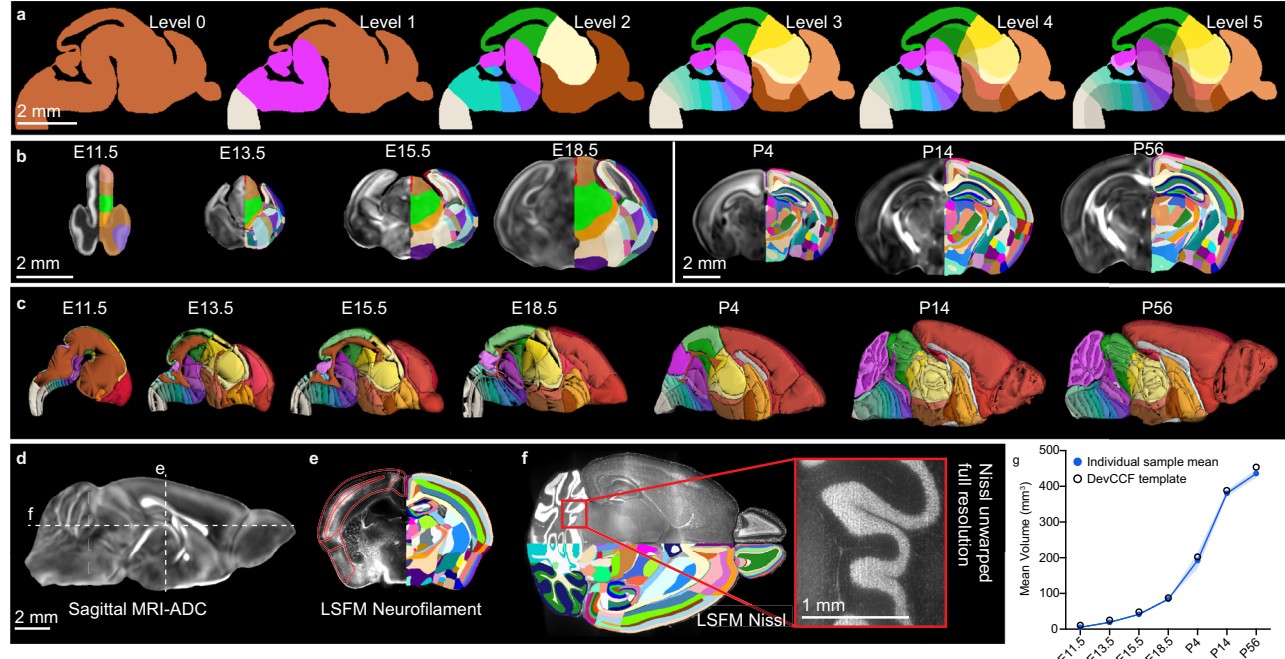

**Fig. 4 | Anatomical segmentations based on a developmental ontology.**
**a** DevCCF ontology levels 0 through 5 displayed over an E15.5 sagittal slice.
**b** Coronal MRI FA slices through the subpallium with anatomical segmentation overlays on the right hemisphere. Embryonic and postnatal brains are depicted at uniform scales, respectively. **c** 3D renderings of DevCCF annotations, not to scale.
**d**–**f** 3D staining guide segmentations, **d** Midline sagittal MRI ADC section indicating location of DevCCF segmentations with aligned neurofilament staining section to guide delineation of cortical areas (red line, **e**) and fluorescence Nissl staining to delineate cerebellar layers (**f**). **g** Brain volume growth curve defined by DevCCF template volume (black open circles; source data in the source data file) and mean ± standard deviation of template generation input sample volumes (blue closed circles and shading; data in Table 2). Scale bars = 2 mm. Annotation colors in **a**, **c** are defined in DevCCF ontology (Supplementary Data 2). Annotation colors in **b**, **d** are semi-randomized to best visualize individual regions.

## Charting early emergence of GABAergic neurons using DevCCF

The DevCCF offers opportunities to map and quantify distinct cell types in developing brains using high-resolution 3D microscopy, as done in adult mouse brains[7,43]. GABAergic cells are excitatory during development and become inhibitory in the mature brain, playing a key role in maintaining excitatory and inhibitory balance[44]. While the developmental origin of cortical GABAergic interneurons is relatively well-studied[45–47], how GABAergic neurons emerge in the whole brain during early embryonic development remains under-studied.

Here, we used *Gad2*-Cre;Ai14 mice that genetically label GABAergic neurons to examine their emergence in developing brains by applying tissue clearing methods and LSFM imaging at E11.5, E13.5, and E15.5 (Fig. 5). We observed tdTomato labeled GABAergic neurons in samples from all three ages (Fig. 5a–c). We registered each sample to the aged-matched DevCCF template to create an averaged image per age (Fig. 5d–g). GABAergic neurons appear as focal clusters in the subpallium (SPall), basal peduncular hypothalamus (PHyB), prosomere 1 (p1), prosomere 3 (p3), and rhombomere 1 basal plate (r1B) as a part of prepontine hindbrain (PPH) by E11.5 (Fig. 5d, g). By E13.5, there is a rapid growth and expansion of GABAergic neurons in each cluster such as the hindbrain (H), including rhombomere 0 (r0) growth from the r1B and the subpallium septum (SeSPall) and striatum (Str) growth from the SPall (Fig. 5e, g). Moreover, GABAergic neurons newly appear in the cerebellar hemisphere (CbH) and midbrain tegmental area (MTg) (Fig. 5e, g). While p1 pretectal and p3 prethalamic areas contain high GABAergic neuron expression, the thalamus in prosomere 2 contains few GABAergic neurons (Fig. 5e, g). By E15.5, there are marked increases of GABAergic cells in all clusters as well as the newly emerging olfactory bulb (Fig. 5f, g). In addition to local expansion of GABAergic neurons, our mapping clearly visualizes chains of migrating cortical interneurons from the SPall to the superficial and deep cortical areas, reaching the half of the neocortex by E13.5 and fully reaching both the neocortex and the hippocampus by E15.5 (Fig. 5h), consistent with previous reports[46,48]. Moreover, we found that migrating interneurons emerge in the intermediate and posterior neocortical areas by E13.5 and reached to the anterior area by E15.5 (Fig. 5h). Using DevCCF segmentations, quantitative analysis revealed the canonical spatiotemporal emergence of GABAergic neurons in developing brains (Fig. 5i).

Hence, we demonstrate that our DevCCF can serve as a standard spatial framework to visualize and quantify spatiotemporal trajectories of fluorescently labeled cell types in developing mouse brains.

## Integrating CCFv3 with P56 DevCCF

The P56 DevCCF provides a unique opportunity to compare developmental ontology-based labels with widely used CCFv3 annotations based on cytoarchitectures[15]. We mapped the CCFv3 labels to the stereotaxically aligned P56 DevCCF template (Figs. 2e and 6a)[14]. While the CCFv3 offers finer segmentation in the isocortex, P56 DevCCF labels provide deeper segmentations in other areas such as the olfactory bulb, the hippocampus, and the cerebellum (Fig. 6a, b). DevCCF isocortical regions have matching cortical layers with CCFv3, but major cortical areas are defined according to those of the ADMBA and the concentric ring topology in the rostrocaudal and mediolateral directions (Fig. 6a, b; Supplementary Fig. 3). We showed the spatial relationship between anatomical labels from each atlas and found several disagreements (Fig. 6b, c). For instance, prosomere 2 (p2) in the diencephalon of the DevCCF annotations overlaps with parts of the hypothalamus (HY), thalamus (TH), and midbrain (MB) in the CCFv3 annotations (Fig. 6b, c). The CCFv3 thalamus encompasses the DevCCF thalamus (p2A; alar plate of p2) and prethalamus (PTh in prosomere 3; p3), while DevCCF pretectum (in prosomere 1; p1) maps to the CCFv3 midbrain (Fig. 6b, Supplementary Fig. 6a, b). Because p1 expresses early diencephalic marker Pax6[49], it belongs to the diencephalon

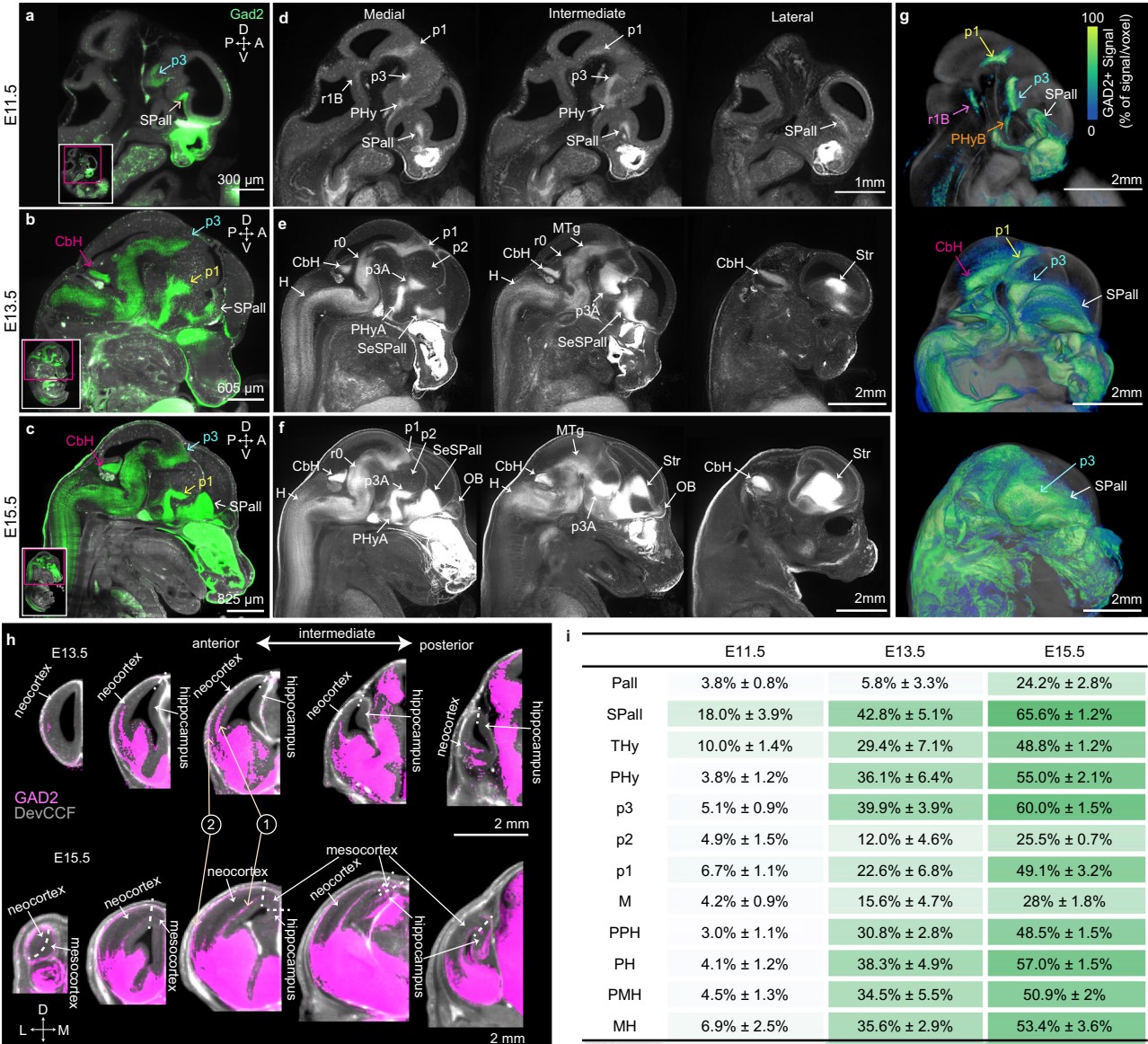

**Fig. 5 | Early emergence of GABAergic neurons in embryonic brains. a** Sagittal images of LSFM imaging from E11.5 *Gad2*-Cre;Ai14 mice show clusters of GAD2+ neurons (green, pseudo colored) in the subpallium (SPall), prosomere 3 (p3). The E13.5 (**b**) and E15.5 (**c**) brains show additional GAD2+ neurons in the cerebellar hemisphere (CbH). Averaged *Gad2*-Cre;Ai14 brains registered to the age matched DevCCF at E11.5 (**d**), E13.5 (**e**), and E15.5 (**f**). The first column is for the medial, the second for the intermediate, and the third for the lateral sagittal planes. **g** Average GAD2+ signals from E11.5 (*n* = 7), E13.5 (*n* = 7), and E15.5 (*n* = 5) brains with 3D rendering overlay on DevCCF templates. Note rapid expansion of GAD2+ neurons at

E13.5 and E15.5 from initial clusters at E11.5. **h** In addition to local expansion, GAD2+ neurons migrate to deep (1) and superficial areas (2) of the pallium to establish cortical interneurons. **i** Quantification of GAD2+ signals (mean % area filled with signal ± population standard deviation) in DevCCF segmentations at E11.5, E13.5, and E15.5 (source data provided in the source data file). Scale bars = 2 mm. CbH cerebellar hemisphere, H hindbrain, MTg midbrain tegmentum, OB olfactory bulb, PHy peduncular hypothalamus, PHyA PHy alar plate, p prosomere, r0 rhombomere 0, r1B rhombomere 1 basal plate, SeSPall septal subpallium, SPall subpallium, Str striatum.

(rather than the midbrain) in the DevCCF (Supplementary Fig. 6c–e)[28,50]. Similarly, within DevCCF neuromeres (including p1, p2, and p3), we see separate segments of the substantia nigra due to early developmental neuromeric segmentations (Supplementary Fig. 6f, g). In contrast, the CCFv3 does not define developmental neuromeres, thereby segmenting the substantia nigra pars reticulata as one midbrain region (Supplementary Fig. 6g).

By integrating existing CCFv3 labels with P56 DevCCF templates, users can decide their choice of anatomical labels to interpret signals of their interest in the same adult DevCCF template space. Further, the two annotations may be used in a common space to complement one another by functionally segmenting regions that are not annotated in either atlas individually, such as the layers of individual cerebellar

lobules (Fig. 6b). We quantified overlapping voxel-to-voxel anatomical label correspondence between DevCCF and CCFv3 to facilitate further comparison of the two anatomical labels with different delineation criteria (Supplementary Data 3).

Leveraging co-registered DevCCF and Allen CCFv3 labels, one can assess data previously mapped to the CCFv3 in a developmental context (Fig. 6d–g). For instance, a study using a spatial transcriptome approach based on Multiplexed Error-Robust Fluorescence in situ Hybridization (MERFISH) identified 34 classes and 338 subclasses of cell types across the whole mouse brain based on CCFv3 ontology (Fig. 6d)[51]. We re-mapped the cell type classification data in the DevCCF and compared the distribution of cell types in the two atlas labels. Mirroring segmentation differences (Fig. 6b, c), cell-type class and

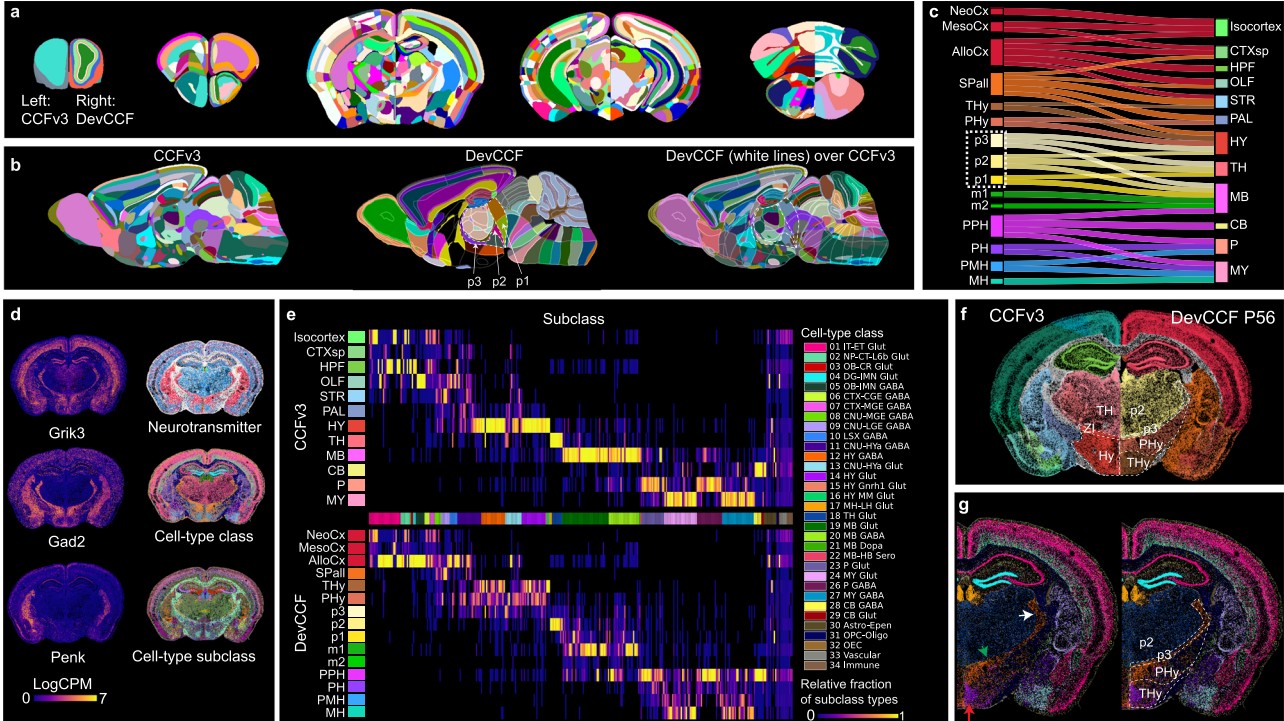

**Fig. 6 | Integrating CCFv3 with DevCCF for enhanced anatomical mapping.**
**a** CCFv3 annotations (left hemisphere) and DevCCF P56 annotations (right hemisphere, https://atlas.brain-map.org/) overlayed on the DevCCF P56 template across five slices from anterior (left) to posterior (right). **b** Sagittal slices with CCFv3 annotations (left, https://atlas.brain-map.org/), DevCCF (middle), and DevCCF label boundaries overlaid on CCFv3 annotations (right). Neuromeric boundaries of the prosomere (p)1, p2, p3 highlighted with white dashed lines over colored CCFv3 annotations demonstrate shared and non-overlapping boundaries of CCFv3 and DevCCF annotations. Annotation colors in **a**, **b** are semi-randomized to differentiate annotations. **c** Sankey diagram illustrates matched structural relationship between DevCCF and CCFv3 annotations. Size of individual areas represents

logarithmic scale of regional volume. **d** MERFISH spatial transcriptome data with three representative genes (left) and cell type classifications (right). **e** Heatmap of cell-type distribution in CCFv3 (top) and DevCCF segmentations (bottom). Heatmap values show proportion of cells in a region relative to the total number of cells in the subclass. Cell-types are ordered by their parent class, denoted by color bars on the x-axis. **f** Registered spatial transcriptome with CCFv3 segmentation (left) and DevCCF segmentation (right). Colors are the same as in **c**. **g** Registered cell type data show the same GABA neurons in the reticular thalamus (Left, white arrow) and zona incerta (Left, green arrow) as a part of the p3 while hypothalamic glutamatergic cell type (Left, red arrow) as a part of terminal hypothalamus (Thy). Source data for plots is provided in source data file.

subclass are distributed differently based on DevCCF labels with developmental ontology. For example, hypothalamic GABAergic cell types (12 HY GABA) are densely located in the reticular thalamus (RT), the zona incerta (ZI), and dorsal hypothalamic area while hypothalamic glutamatergic neurons (14 HY Glut) clustered in the ventral hypothalamus in the CCFv3 (Fig. 6f, g). In contrast, both the RT and the ZI are considered as the p3, a part of the diencephalon based on the developmental ontology, which is additionally supported by similar cell type distribution of GABAergic neurons (Fig. 6f, g). Moreover, DevCCF divides the hypothalamus into peduncular hypothalamus (PHy) and terminal hypothalamus (THy), which is populated by 12 HY GABA and 14 HY Glut, respectively. Similarly, the DevCCF considers the cerebellum as a part of the prepontine hindbrain (PPH) rather than a separate structure (as in CCFv3) because the cerebellum emerges in the rhombomere 0 and 1 as a part of the PPH during development (Fig. 6c, e)[52].

Hence, DevCCF offers opportunities to re-interpret brain areas and emerging spatial genomic data in the context of developmental ontology.

### Web visualization for DevCCF
We created a Neuroglancer v3.5 based web interface (https://kimlab.io/brain-map/DevCCF/) to interactively visualize and explore the DevCCF. The platform features co-registered MRI and LSFM templates as well as DevCCF annotations at the seven developmental ages. Additionally, the P56 DevCCF includes co-registered CCFv3 annotations to compare

with DevCCF annotations. Users can digitally explore 2D slice viewers that may be freely rotated to view in any orientation. Template and annotation datasets may be displayed or hidden as independent layers and modified with user defined color, contrast, and opacity (Fig. 7a, b). Hovering over a region reveals its name, abbreviation, and assigned numerical identifier. Region selection reveals additional information, such as region abbreviation, subregions, and parent regions in the ontology tool (Fig. 7c). Users can apply these interactive tools to create layer structures that highlight specified anatomical features in a single template (Fig. 7d–f) or simultaneously explore features of multiple overlayed or side-by-side templates to gain additional anatomical context in a single viewer (Fig. 7g–i).

### Discussion
Here, we present the DevCCF as a set of 3D atlases for the developing mouse brain including multi-contrast MRI and LSFM templates and corresponding 3D annotations based on a developmental ontology. The DevCCF can facilitate the interpretation and integration of diverse data across modalities and scales including advanced high resolution 3D imaging and spatial genome data[20,53].

The DevCCF contains undistorted morphology and intensity averaged symmetric templates at seven key developmental ages: E11.5, E13.5, E15.5, E18.5, P4, P14, and P56. Stereotaxically aligned postnatal templates provide spatial coordinates to guide surgical procedures (e.g., stereotaxic brain injection). Template ages correspond with the ADMBA and cover critical stages of regional differentiation[28]. Across 7

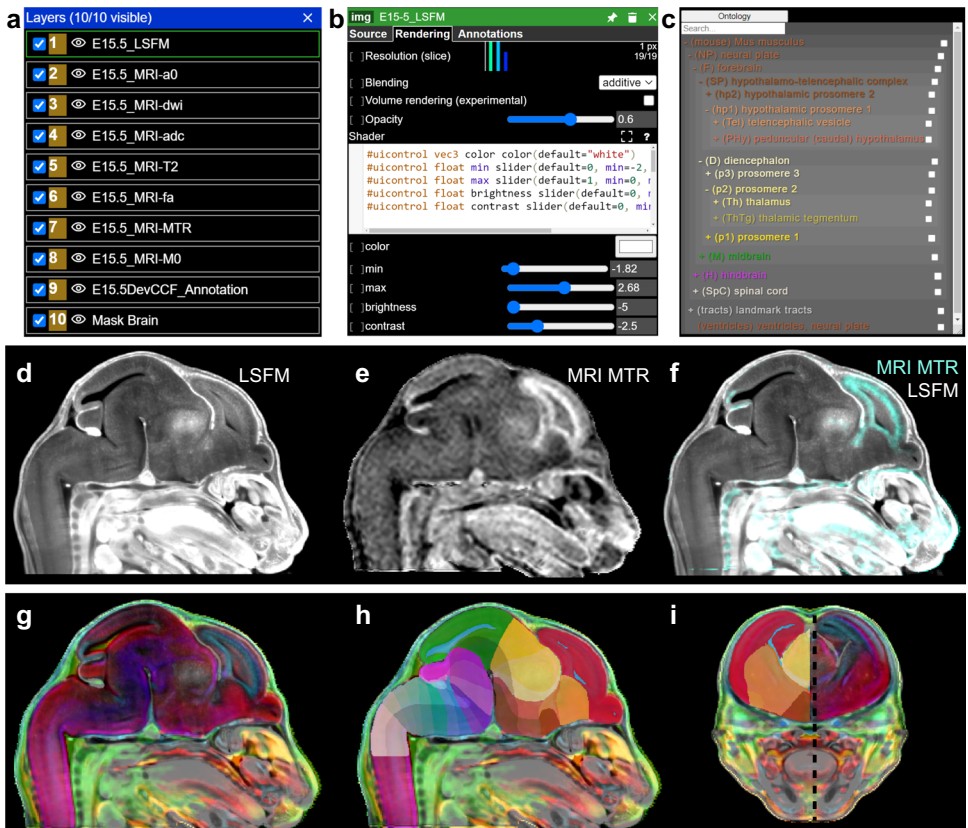

**Fig. 7 | Web Visualization for DevCCF. a** Layer panel allows layer selection with a right click and hiding by selecting the eye icon to the left of the layer. **b** Once selected, the layer edit tool enables user modification of layer color, contrast, and brightness. **c** The ontology viewer allows search and selection of individual segmentations and parent regions by ontological layer. When a region is selected in the viewer or ontology tool, the region's metadata is displayed including region name, abbreviation, and ID. The ontology tool may be dragged to any location in the viewer. Neuroglancer allows users to visualize either individual E15.5 LSFM autofluorescence in **d**, MRI MTR template in **e**, or their overlay **f**. E15.5 DevCCF templates overlaid (**g**) and segmentation (**h**, **i**). Black dashed line in **i** denotes sagittal slice location of **g**, **h**. Annotation colors defined in DevCCF ontology (Supplementary Data 2).

ages, the DevCCF contains 136 total samples (66 MRI, 70 LSFM), which is then doubled by flipping each sample for symmetry. Some developmental MRI atlases approach DevCCF sample sizes, but are restricted to MRI modality with approximately half DevCCF resolution and limited segmentations at each age (Table 1)[34,37]. The 3D-reconstructed ADMBA meets DevCCF resolution and breadth of ages, but is based on a single 2D sectioned sample[33]. In contrast, DevCCF contains average templates with sharper contrast that can represent overall morphology better than individual samples alone. Moreover, the DevCCF contains at least four MRI contrast templates along with an aligned age-matched LSFM autofluorescence template, which helps to register data from different modalities[20,54,55].

To test the DevCCF, we mapped whole brain LSFM imaging, spatial transcriptome, and ADMBA ISH data to various DevCCF ages, demonstrating how researchers can easily align data from different imaging methods to the DevCCF. For instance, we mapped the emergence of GABAergic neurons in the whole embryonic brain using LSFM imaging, demonstrating that the DevCCF is a robust tool to enable precise spatial localization of cell type data and promote comparative analyses across diverse datasets through development[10,56–58].

Although distinct cytoarchitecture (e.g., Nissl visualization) has been useful to segment adult brain atlases[12,13], it is difficult to apply in developing brains due to immature and rapidly migrating cells. The DevCCF ontology is adapted from ADMBA labels[28], which utilize an original developmental mouse brain ontology with up to 13-level hierarchical ontology based on the prosomeric model to outline genetically driven vertebrate brain development[28,32]. Our adapted annotations account for contemporary developmental anatomy,

including the notable re-organization of the pallium into a concentric ring topology[39,42]. The expert guided 3D digital annotations allow users to examine labels with any desired angle and serve as a neuroinformatic tool to quantify signals across the developing mouse brain[15,59–61].

Inconsistencies in anatomical segmentation between different atlases have caused confusion within the community[14]. We observed partially overlapping, yet distinct borders with varying depth of segmentations between community standard CCFv3 (mostly based on adult cytoarchitecture) and P56 DevCCF. Therefore, we aligned the CCFv3 template and annotations to P56 DevCCF templates to place the two atlas annotations within the same template. This allows users to toggle between CCFv3 and DevCCF segmentations on our interactive web platform, providing information on how CCFv3 defined areas can be reclassified in DevCCF based on the developmental origin. We also show that transcriptionally identified subclasses are concordant with anatomical boundaries in both atlases. For example, MERFISH derived transcriptional subclasses[51] of the DevCCF PPH align with two distinct parent regions in CCFv3: the cerebellum and part of the pons. While the cerebellum appears cytoarchitecturally distinct from the pons by adulthood, this corresponds with our knowledge about the development of the medial cerebellar vermis from the isthmic rhombomere (r0) and the lateral cerebellar hemisphere from rhombomere 1[42], and with our PPH growth curve confirming delayed growth as in previous literature[34]. We argue that neither ontology is superior. Rather, an ontology should be selected based on user interest and experimental questions (e.g., adult cytoarchitecture vs developmental origin).

We envision that the DevCCF will be widely used to support developmental and adult mouse brain mapping experiments.

Advances in tissue clearing and high resolution microscopy have been critical to increase our ability to examine individual cell types in the whole mouse brain at single cell resolution[18,61–63]. MRI has been used to examine macroscopic changes in brain morphology, connectivity, and function with well-established registration methods in humans and other species[29,64–71]. Furthermore, recently developed spatial transcriptomic approaches have begun to unravel the gene expression landscape in the adult and developing mouse brains[8,10,11,51,72–75]. For instance, the molecular atlas of the adult mouse brain utilized spatial transcriptomics to identify brain areas with similar molecular profiles and created anatomical labels with automated segmentations[8]. When integrated and compared to the DevCCF P56 template, we found both shared and diverging segmentations due to different delineation criteria (Supplementary Fig. 7). The DevCCF can serve as a resource to map and integrate the growing quantity of developmental mouse brain spatial transcriptomics data[11,58] and facilitate interoperability between different studies. Once integrated, these large multimodal datasets can be used with computational tools to unveil complex developmental mechanisms and drive automated segmentation[8] at a scale not possible by human eye alone.

The DevCCF is not without limitations. Increasing sample size and enhancing registration methods could improve template signal contrast, enabling updates to multimodal alignment and anatomical parcellation, and improving experimental data alignment[15,55]. Increasing sample size will be possible as we and others continue to share 3D whole brain data via open repositories, such as OpenNeuro and Brain Imaging Library[56]. Further registration enhancements will possible using future forms of registration tools such as Elastix[76], FSL[77], and ANTs[38]. This will also improve multimodal registration,[23,24,54,55] and inclusion of additional 3D imaging modalities such as micro-computed tomography, which could be used to expand the DevCCF's utility[23,24,78]. As postnatal templates do not include skull data, stereotaxic coordinates are defined by anterior commissure and estimated bregma, reducing the accuracy anterior-posterior alignment during in vivo stereotaxic procedures. Adding skull data to postnatal templates will enable lambda and bregma defined stereotaxic coordinates, improving alignment for such procedures. On the user end, expansion of multimodal atlas registration software such as mBrainAligner-Web[22,23] and BrainGlobe[79] to include the DevCCF would simplify experimental data integration. Future refinement of anatomical segmentations is warranted including white matter tracks. We also expect that experimental discoveries and automated segmentation based on diverse mapped data will necessitate further segmentation and updates to DevCCF annotations[80,81]. Moreover, the selected seven developmental ages, though comprehensive, may be too sparse to fully capture the rapid cellular changes occurring in early brain development. Cross-age registration and interpolation can generate mappings between template ages that characterize morphological changes and examine aligned datasets over the temporal axis[29]. Addressing the DevCCF gap in early postnatal development, we have shared the early postnatal developmental mouse brain atlas with 3D templates at P4, 6, 8, 10, 12, and 14[82].

Importantly, grounded in the prosomeric model of vertebrate development, the DevCCF can serve as the foundation for constructing developmental atlases for other mammalian species (e.g., macaque, human)[83–85]. Therefore, future efforts should focus on continued refinement of the DevCCF and analogous atlases in diverse species. We conclude that the DevCCF establishes a standardized anatomical framework for investigating developing mouse brains and facilitates collaborative and reproducible advancements in neuroscience research.

## Methods
### Animals
All experiments and techniques involving live animals have been approved and conform to the regulatory standards set by the Institutional Animal Care and Use Committee (IACUC) at the Pennsylvania State University College of Medicine. We used in-house bred C57bl/6 J mice (originally purchased from the Jackson Laboratory, Strain #:000664) or transgenic animals with C57bl/6 J background to create MRI and LSFM templates. For brain-wide labeling of pan-GABAergic cell types during embryonic and early postnatal development, we used Gad2-IRES-Cre mice (The Jackson Laboratory, stock 028867) crossed with Ai14 mice which express a Cre-dependent tdTomato fluorescent reporter (The Jackson Laboratory, stock 007908). We used tail samples with PCR for genotyping including Rbm31-based sex genotyping for mice younger than P6. All mice were housed under a 12-hour light/12-hour dark cycle at 22–25 °C with access to food and water ad libitum.

### Timed pregnancies and brain sample collection
Timed pregnancies followed recommendations from The Jackson Laboratory (https://www.jax.org/news-and-insights/jax-blog/2014/september/six-steps-for-setting-up-timed-pregnant-mice). Adult breeder males were singly housed for 1 week prior to pairing. We paired a male and a female breeder in the evening, removed the male breeder the following morning, and checked for the presence of a vaginal plug in the females. After separation, we measured the baseline body weight of all females on what we considered E0.5 and added extra enrichment to the cages for female breeders. Subsequently, we weighed the females again on E7.5 and E14.5 to assess weight gain, expecting a 1 g or 2–8 g gain at these respective timepoints. Pregnancy was confirmed either by the presence of fetuses during euthanasia of the dams and fetal tissue collection (on E11.5, E13.5, E15.5, and E18.5) or by the birth of a litter.

On target collection dates for embryonic samples (E11.5, E13.5, E15.5, and E18.5), pregnant dams were placed in an isoflurane chamber until deeply anesthetized at which point a mixture of ketamine and xylazine was administered via intraperitoneal injection. The dams were subjected to cervical decapitation once fully anesthetized. Then, the uterine horns were immediately removed and placed in ice-cold petri dishes filled with 0.05 M PBS for careful removal of embryos from the uterine casing. E18.5 brains were dissected out at this point while E11.5, E13.5, E15.5 were processed as whole embryos. Embryonic samples were incubated in a 4% paraformaldehyde (PFA) in 0.05 M PBS solution for two days at 4 °C before storing in 0.05 M PBS at 4 °C until use. All embryos were characterized according to the Theiler Staging (TS) Criteria for Mouse Embryo Development[86]. We used E11.5 for TS19, E13.5 for TS21, E15.5 for TS24, and E18.5 for TS26. After PFA fixation was complete, dissection of the embryos from the yolk sac (E11.5) and eye/eye pigment removal (E11.5, E13.5, E15.5) were performed under a dissection microscope (M165 FC Stereomicroscope, Leica), followed by storage in 0.05 M PBS at 4 °C until use. Whole embryos were used for E11.5, E13.5, and E15.5, and dissected brains were used for E18.5 for tissue clearing and LSFM imaging. All embryonic samples were tailed for sex genotyping.

For postnatal brains (P4, P14, and P56), we defined pups at birth as P0. For collection, mice were anesthetized by a mixture of ketamine and xylazine via intraperitoneal injection. Anesthetized animals were subsequently perfused by saline (0.9% NaCl) and freshly made 4% PFA. Decapitated heads were fixed in 4% PFA at 4 °C overnight, followed by brain dissection and storage in 0.05 M PBS at 4 °C until use.

The Kim Lab at Penn State University prepared all animal samples and performed tissue clearing with LSFM imaging. Whole embryos (E11.5, E13.5, E15.5) or decapitated samples (E18.5, P4, P14, and P56) were sent to Dr. Jiangyang Zhang's lab at NYU for high resolution MRI.

### Magnetic resonance imaging
All imaging was performed on a horizontal 7 Tesla MRI system (Bruker Biospin, Billerica, MA, USA) equipped with a high-performance gradient system (maximum gradient strength of 670 mT/m). We used a

transmit volume coil (72 mm inner diameter) together with a 4-channel receive-only phased array cryoprobe with high sensitivity. Hair and scalp were removed and heads with intact skulls were imaged to prevent deformations. As the embryonic mouse has immature soft skulls, embryonic mouse heads were embedded in 5% agarose gel (Sigma Aldrich, St Louis, MO, USA) for additional support. Specimens were placed in 5 mL syringes filled with Fomblin (Solvay Solexis, Thorofare, NJ, USA) to prevent dehydration and susceptibility to artifacts.

High-resolution diffusion MRI was acquired using an in-house 3D diffusion-weighted gradient and spin-echo (DW-GRASE) sequence[87] with the following parameters: echo time (TE)/repetition time (TR) = 30/400 ms, two signal averages, diffusion gradient duration/separation = 4/12 ms, 60 diffusion directions with a b-value of 1.0 ms/$\mu$m$^2$ for E11.5, 2.0 ms/$\mu$m$^2$ for E13.5-E17.5, and 5.0 ms/$\mu$m$^2$ for P4, P14, and P56 brains. The increase in b-values with age was necessary as the diffusivity of brain tissues decreases with development[88]. Co-registered T2-weighted data were acquired using the same sequence but with TE/TR = 50/1000 ms. The native and interpolated spatial resolutions of the MRI data were 0.063/0.0315 mm for E11.5, 0.068/0.034 mm for E13.5, 0.075/0.037.5 mm for E15.5, 0.08/0.04 mm for E18.5, and 0.1/0.05 mm isotropic for P4-P56.

The 3D MRI data were reconstructed from k-space to images and zero-padded to twice the raw image resolution in each dimension in MATLAB R2022a (Mathworks, Natick, MA, USA). Diffusion tensor images[89] were constructed using the log-linear fitting method in DTI Studio v1.8 (http://www.mristudio.org), and the tensor-based scalar metrics were generated, including the mean diffusivity (MD) and fractional anisotropy (FA).

## Tissue clearing

We mainly used SHIELD (Stabilization under Harsh conditions via Intramolecular Epoxide Linkages to prevent Degradation) tissue clearing to ensure minimal tissue volume changes while preserving endogenous fluorescence signals when available[90]. Commercially available SHIELD preservation, passive clearing reagents (see Supplementary Data 4), and detailed protocols were obtained from LifeCanvas Technologies (https://sites.google.com/lifecanvastech.com/protocol/outline). For P56 brains, PFA-fixed samples were incubated in SHIELD OFF solution for 4 days at 4 °C on an orbital shaker. Subsequently, the SHIELD OFF solution was replaced with SHIELD ON buffer and incubated for 24 hours at 37 °C with shaking. Tissues were incubated in 20 mL of delipidation buffer at 37 °C for 4-6 days followed by an overnight wash in 1x PBS at 37 °C with gentle shaking. If sample imaging did not occur within a week after the delipidation step, tissue samples were stored in 1x PBS containing 0.02% sodium azide at 4 °C before continuing to the next step. To match the refractive index (RI) of the delipidated tissues (RI = 1.52) and obtain optical clearing, samples were incubated in 20 mL of 50% EasyIndex (LifeCanvas Technologies, Cat. no.: EI-Z1001) + 50% distilled water for 1 day, then switched to 100% EasyIndex solution for another day at 37 °C with gentle shaking. For samples in earlier time points, we used whole embryos (E11.5, E13.5, and E15.5) or dissected brains (E18.5, P4 and P14) with the same protocol but with smaller reagent quantities and shorter incubation times. Once cleared, embryonic samples were stored in tightly sealed containers with 100% EasyIndex at room temperature (20-22 °C). For 3D immunolabeling and histological staining, we used electrophoresis based active clearing using the SmartClear II Pro (LifeCanvas Technologies) and active labeling using SmartLabel (LifeCanvas Technologies) based on LifeCanvas protocols. Briefly, after the SHIELD OFF step described above, samples were incubated in 100 mL of delipidation buffer overnight at room temperature (RT). Samples were inserted in a mesh bag, placed in the SmartClear II Pro chamber, and delipidated overnight. Samples were transferred to 20 mL of primary sample buffer and incubated overnight at RT. Samples were placed in a sample cup in the SmartLabel device with an antibody cocktail to perform active labeling[91]. We used Mouse Monoclonal Antibody Neurofilament NF-H (Encor, cat. no. MCA-9B12, Lot no. 012022, 10 $\mu$l diluted 1:100 per hemisphere) with Alexa Fluor® 647 AffiniPure™ Fab Fragment Donkey Anti-Mouse IgG (H + L) secondary antibody (Jackson Immuno Research, cat. no.: 715-607-003, Lot no. 153490, 2.7 $\mu$l diluted 1:500 per hemisphere), and propidium iodide (Thermo Fisher, cat. no.: P1304MP, 12 $\mu$l /hemisphere) for pan-cellular labeling. For the limited dataset at P56, we used the iDISCO (immunolabeling-enabled three-dimensional imaging of solvent-cleared organs) tissue clearing protocol[92,93].

## Light sheet fluorescence microscopy imaging and reconstruction

For LSFM imaging, all samples were embedded in an agarose solution containing 2% low-melting agarose (Millipore Sigma, cat. no.: A6013, CAS Number: 9012-36-6) in EasyIndex using a custom sample holder. Embedded samples were then incubated in EasyIndex at room temperature (20-22 °C) for at least 12 hours before imaging using the SmartSPIM light sheet fluorescence microscope (LifeCanvas Technologies, Cambridge, MA, USA). During the imaging process, the sample holder arm securing the embedded sample was immersed in 100% EasyIndex. Our imaging setup consisted of a 3.6X objective lens (LifeCanvas Technologies, 0.2 NA, 12 mm working distance, 1.8 $\mu$m lateral resolution), three lasers with wavelengths of 488 nm, 561 nm, and 642 nm, and a 5 $\mu$m z step size. After imaging, all samples were stored in 100% EasyIndex at room temperature (20–22 °C). For our iDISCO cleared P56 brain samples, we did not use agarose embedding and directly mounted samples in a custom-built holder. LSFM imaging was performed in ethyl cinnamate for index matching (Millipore Sigma, cat.no.: 112372, CAS number: 103-36-6) using the same imaging parameters.

For 3D reconstruction, we developed a parallelized stitching algorithm optimized for conserving hard drive space and memory consumption initially based on Wobbly Stitcher[17]. The algorithm started by collecting 10% outer edges of each image tile and making a maximum intensity projection (MIP) of outer edges in the axial (z) direction for every set of 32 slices of the entire stack. The algorithm then aligned z coordinates of MIP images across image columns, followed by the x and y coordinate alignment. Finally, 32 slices within each MIP were adjusted based on curve fitting to reach final coordinates of each tile. This algorithm only reads the raw images two times (at the beginning and the final writing), which significantly reduced the bottleneck of reading large files in a storage drive. See Code Availability section for algorithm access.

## Symmetric template construction

Each symmetric template is an intensity and morphological average of multiple male and female samples with a sample size ranging from 6 to 14 (Table 2). After stitching, images were preprocessed for template construction. For postnatal ages, MRI data preprocessing began with digital brain extraction by hand drawing brain masks using ITK-SNAP v4[94] or Avizo 2021.1 (Thermo Fisher Scientific). Brain tissue was assigned a value of 1 and non-brain was assigned 0. The mask was multiplied by MRI data to extract the brain. Next, sample orientation was viewed in ITK-SNAP and corrected using ANTs v2.3.5[38]. MRI data was normalized using N4 bias field correction[95]. LSFM data preprocessing began with image resampling to 3 sizes: 50 $\mu$m, 20 $\mu$m, and 10 $\mu$m isotropic voxel resolution, then orientation correction. To ensure template symmetry, each preprocessed image was duplicated and reflected across the sagittal midline, doubling the number of input datasets used in the template construction pipeline and allowing final parcellations to be reflected over the midline, reducing annotation efforts. Template construction, ANTs call 'antsMultivariateTemplateConstruction2.sh'[38,96], was employed on Penn State's High-Performance Computing system (HPC). Briefly,

 

starting from an initial template estimate derived as the average image of the input cohort, this function iteratively performed three steps: (1) non-linearly registered each input image to the current estimate of the template, (2) voxel-wise averaged the warped images, and (3) applied the average transform to the resulting image from step 2 to update the morphology of the current estimate of the template. Iterations continued until the template shape and intensity values stabilized. MRI templates were constructed at their imaged resolution using ADC MRI contrasts for initial postnatal templates and diffusion weighted imaging (DWI) contrasts for embryonic templates. Once the initial MRI template was constructed, the sample to template warp fields generated were applied to all MRI contrasts for each sample. Warped samples were averaged to construct templates for each contrast. LSFM templates were constructed from autofluorescence data collected from C57bl/6 J mice and transgenic mice with a C57bl/6 J background. To save memory and improve speed, LSFM templates were initially constructed at 50 μm isotropic resolution. This template was resampled for template construction initialization at 20 μm isotropic resolution, a process repeated to construct the final LSFM template with 10 μm isotropic resolution input images. The in-house template construction script is referenced in the Code Availability section.

### Multimodal registration of 3D imaging to the DevCCF

We aligned the CCFv3 to the P56 DevCCF and each LSFM template to the corresponding age matched DevCCF MRI template to enable data integration across multiple modalities with undistorted morphology. Our protocol aims to address multimodal registration challenges due to differences in brain and ventricle volume that often result in internal structure misalignment[97]. We performed initial non-linear registration of the 3D datasets (CCFv3 and LSFM templates) to the age-matched DevCCF MRI template using ANTs with the mutual information similarity metric[38]. We then visually compared the warped 3D dataset with the DevCCF template in ITK-SNAP[98] to identify landmark brain regions that remained misaligned after the initial registration. Whole brain masks and misaligned regions were segmented for the 3D dataset and DevCCF template in 3D using Avizo (Thermo Fisher Scientific). The segmented regions were subtracted from the brain masks, creating modified brain masks with identifiable boundaries around misaligned brain regions. This provided a map of regions that needed correction. Next, linear registration was performed of the modified brain masks, followed by equally weighted non-linear alignment of both the 3D data images and their modified brain masks. Landmark-assisted multimodal registration warp fields were resampled and applied to transform CCFv3 and LSFM templates to MRI template morphology at 20 μm isotropic voxel resolution, which is sufficient resolution for mapping cell-type data with a reasonable compute time, yet small enough file size to be downloadable to a local computer. CCFv3 annotations were also transformed to MRI template morphology at 20 μm isotropic voxel resolution for comparison with the DevCCF. We elected not to register 10 μm isotropic voxel resolution images to MRI for several reasons Increasing resolution does not have sufficient resolution in MRI with which to be paired, therefore is unlikely to improve resolution. Additionally, computation time and resources to register the 10 μm isotropic voxel resolution was not reliably available. Lastly, this helps keep the DevCCF package small enough to easily download and work with on any computer. Because 10 μm isotropic voxel resolution LSFM templates were generated, we have made them available unwarped alongside the DevCCF.

We also registered LSFM whole brain data featuring immunostaining (e.g. Nissl) or Cre-dependent fluorescence (e.g. *Gad2*) to the DevCCF to assist with annotation, segmentation, and validation. Each LSFM dataset collected up to three channels simultaneously, always including one autofluorescence channel. We used non-linear registration to align LSFM autofluorescence channel data to the warped DevCCF LSFM template. The LSFM template was chosen to achieve optimal registration quality due to matching the autofluorescence contrasts. Forward transforms were applied to LSFM immunostaining and Cre-dependent fluorescence channel data to align them to the DevCCF template.

### 2D gene expression mapping onto DevCCF

We utilized the Allen Brain Atlas API to download in situ hybridization (ISH) data from the ADMBA[28] (E11.5, E13.5, E15.5, E18.5, P4, P14, P28) and Allen Brain Atlas[50] (P56) as both imaged histological sections and analyzed gene expression. Except for P28, this dataset is age-matched to the DevCCF, which allows reliable validation of DevCCF annotations. Each dataset consists of 2D images of coronal or sagittal sliced brain sections depicting expression of a single gene. Each brain sample was used for 4 to 7 ISH gene expression experiments, alternating slices for each.

We used the python v3.8.16 package ANTsPy v0.3.8[38] to register the ISH data to the age-matched DevCCF. P28 data was matched to the P56 DevCCF as mouse brain volume is stable after 3 weeks of age[27]. To prepare for registration, sample level data was compiled from each of 4-7 ISH experiments, then reconstructed to a 3D volume containing all sample sections and 3D volumes of each individual analyzed gene expression experiment. Missing sections were included in the volumes as empty slices. Reconstructed volumes were resampled to 512×512 pixels for computational tractability and histological reconstruction intensity was inverted to set the background to zero. Empty histological analyzed sections were filled using B-spline scattered data approximation[99] to generate a full intact volumes.

To begin alignment, we ran an initial linear multi-metric registration from DevCCF MRI templates (FA and ADC contrasts) to the reconstructed histological sample volume. This aided the next step of slice-wise correction of the 3D sample reconstruction. Here, we ran nonlinear multi-metric registration of each sample section to the neighboring experimental section and the aligned MRI template section mask. Next, the template to subject alignment was refined by nonlinearly registering the age-matched MRI templates to the slice-wise corrected sample reconstruction. Finally, the B-spline approximation filled analyzed gene expression volumes for each individual experiment were warped to DevCCF morphology using the transform parameters saved during sample slice-wise correction and inverse transform parameters saved during refined template to subject alignment. All registrations are used the ANTsPy 'antsRegistrationSyNQuick' transform type with the mutual information similarity metric, with the exception of slice-wise correction using the MRI mask, which used the mean squared difference similarity metric[38].

### Integrating CCFv3 to DevCCF

To compute the voxel-to-voxel mapping between CCFv3 and DevCCF anatomical label correspondence, we performed voxel-wise comparison in the P56 aligned annotations. Each voxel in the P56 brain was labeled by one DevCCF and one CCFv3 annotation and grouped into a parent structure. Unique combinations of DevCCF-CCFv3 mappings were summed to generate voxel counts. Voxel counts meeting a threshold of 250 voxels were then plotted as a Sankey flow diagram using the Plotly v5.10 library in python 3.11.

To quantify cell type distribution between CCFv3 and P56 DevCCF labels for spatial transcriptomic data, we obtained soma coordinates for each cell in the MERFISH dataset in CCFv3 morphology[51]. We used the CCFv3 to DevCCF inverse transforms to warp the P56 DevCCF annotations to the CCFv3 template. For each coordinate, we compared the DevCCF and CCFv3 labels and grouped cells by parent division and subclass label. We then normalized the data by the total cell count per subclass, resulting in a proportional representation of cell subclasses by parent division. Heatmaps were plotted using python's seaborn 0.12 library. Code associated with these methods are referenced in the Code Availability section.

Molecular Atlas[8] 3D meshes for each region were downloaded from molecularatlas.org and pixelized using the Dragonfly 2022.2 (Comet Technologies Canada Inc., Montreal, Canada) multi ROI toolbox, giving the Molecular Atlas in CCFv3 space. The Molecular Atlas was transformed to DevCCF morphology using CCFv3 to DevCCF inverse transforms for annotation comparison.

### Anatomical segmentations for the DevCCF

We performed theory and data-driven anatomical segmentation of the multimodal templates at each age using Avizo (Thermo Fisher Scientific). We manually drew contours on coronal, horizontal, and sagittal slices of templates to generate 3D segmentations. To assist in the process, we used various interpolation, thresholding, and smoothing tools. We assigned unique labels and colors to each region and further developed the hierarchical nomenclatures following standards in the ADMBA. Annotations were delineated on one hemisphere, then copied and reflected over the midline to the empty hemisphere to make them bilateral.

We followed the principles of the prosomeric model to define brain regions based on morphological features, such as sulci, fissures, ventricles, commissures, tracts, cytoarchitecture, and gene expression[32]. Segmentations started with large regions defined early in development, such as the neural plate, and were progressively subdivided into smaller regions as defined by the prosomeric model, such as the forebrain, hindbrain, and spinal cord until reaching a level of detail comparable to at least level 5 (fundamental caudo-rostral and dorsoventral partitions) in all brain regions (Fig. 4a). Additionally, we updated the telencephalon to reflect the cortical concentric ring topology (Supplementary Fig. 3)[39]. Anatomical divisions and subdivisions were drawn across the whole brain in multiple segments and combined. For example, floor, basal, alar, and roof plate segmentations in the ventral-dorsal direction were drawn separately from rostro-caudal neuromere segmentations and later overlayed with one another (Table 3). Similarly, cortical region segmentations (e.g., hippocampal cortex and olfactory cortex) were drawn separately from cortical layer segmentations (e.g., ventricular zone, mantle zone), and later overlayed to combine. Likewise, the subpallium was segmented by the primary domains (striatum, pallidum, diagonal, and preoptic) and secondary divisions (septum, paraseptum, central, and amygdalar), then combined. This technique allows efficient segmentation correction upon validation. Additional brain regions were modified from the ontology described in the ADMBA to meet the current anatomical understanding (Supplementary Data 2)[39,42,100].

Neuromeres and pallial cortex division boundaries follow smooth trajectories that end perpendicular to the brain surface and the ventricles. Each neuromere stretches from floor to roof plate and is in contact with only one neuromere caudally and one neuromere rostrally[32]. Pallial neocortex (NeoCx), allocortex (AlloCx), and mesocortex (MesoCx) are segmented according to the concentric ring topology as 6-layer structures, 3-layer structures, and transitional structures, respectively, where layers are overlayed from ventricular to superficial (Supplementary Fig. 3)[39]. Choroidal tissue is within the roof plate. The medullary hindbrain (MH) neuromeres are divided evenly into 5 segments (r7-r11). Parcellations are based on previously defined landmarks[28,32,39,42,100–102] primarily visualized with template contrast features. For example, the lateral and medial ganglionic eminences mark the early subpallium (Supplementary Fig. 4a). The facial motor nerve identifies the ventral surface and alar-basal boundary of rhombomere 6 (r6) (Supplementary Fig. 4e). In late postnatal ages, the neocortex, allocortex, and mesocortex were segmented by separating regions by thick cortical layer areas, 3-layer areas, and transitional zones, respectively, visible in MRI templates (Supplementary Fig. 4f, h). Regions without easily visible landmarks are delineated based on neighboring regions. Mesomere 2 (m2), for example is a thin neuromere between the caudal most landmarks of mesomere 1 (m1) and the rostral most landmarks of isthmus (r0) (Table 3).

The morphological foundation of the DevCCF was primarily guided by MRI templates, additional details from warped LSFM autofluorescence templates and registered LSFM cell-type data (e.g. Neurofilament, Nissl, *Gad2*). Here we provide a few examples of each data type (Figs. 3, 4d–f, Table 3). The MRI T2-weighted templates, in conjunction with LSFM templates, facilitated delineation of brain tissue from non-brain structures, as well as ventricle identification in embryonic DevCCF templates (Fig. 3). MRI FA templates highlight white matter tract landmarks, which serve as markers for many boundaries (e.g., the boundary of p1 and p2 is immediately caudal to the retroflex tract; (Fig. 3p, Table 3). LSFM templates were instrumental in segmenting features such as the choroid plexus in the developmental roof plate (Fig. 3m, q). We imaged and aligned additional 3D fluorescent cell-type specific datasets to provide reference data for DevCCF segmentations. For example, we used LSFM images of embryonic *Gad2*-Cre;Ai14 samples to delineate the subpallium (SPall), prosomere 1 (p1), prosomere 3 (p3), and the cerebellum (Fig. 5). Additionally, LSFM imaging of SYTOX stained mouse brains samples provided Nissl-like 3D histology, allowing us to use classical neuron cell density patterns during segmentation (Fig. 4f).

The ADMBA and associated ISH data were key resources to segment large rostro-caudal, and dorsoventral boundaries. While drawing segmentations, we viewed the atlas and gene expression data side-by-side via the web-portal (https://developingmouse.brain-map.org/) as supportive evidence for the prosomeric model of vertebrate brain development, aiding in validating relative relationships among neuromeres, dorsoventral plates, nuclei, white matter tracts, and cranial nerve roots. Moreover, we utilized registered ISH data associated with the ADMBA to validate DevCCF segmentations based on gene expression patterns known from existing literature (Supplementary Fig. 4).

The CCFv3[15] provided delineations of the cortical layers and many nuclei in the P56 mouse brain. The CCFv3 template and annotations were aligned to the P56 MRI template using our landmark-assisted multimodal registration methods. Previously validated CCFv3 cortex layers (1, 2/3, 4, 5, 6a, and 6b) were imported to the P56 DevCCF. P56 DevCCF cortical regions (e.g., insular cortex, entorhinal cortex) were segmented manually and combined with imported CCFv3 layers to finalize cortical segmentations. Additional CCFv3 segmentations with corresponding prosomeric model regions (e.g., subdivisions of prosomere 2 and mesomere 1) were aligned to the DevCCF and used as primers to segment the DevCCF. CCFv3 cortical layers were also used for P14 and P56. CCFv3 cortical layers were similarly aligned from the P56 DevCCF to the P14 and P4 DevCCF templates via landmark-assisted multimodal registration methods and used to prime segmentation.

Anatomical segmentations were paired with numerical identifiers (IDs). IDs below 18000 represent previously existing ADMBA labels with potential minor name, abbreviation, and ontology level updates. Newly added structures to the DevCCF ontology have IDs in the range 18000-19999. To ensure compatibility with viewing and analysis tools requiring 16-bit depth labels, existing 32-bit ADMBA IDs that fell outside the 16-bit depth range (0-65535) were fit into the 20000 to 29999 range. This was achieved through a formulaic approach where the last four digits of the original 32-bit IDs were extracted and given a prefix of 2. New 16-bit labels ensure the modified IDs retain their uniqueness and do not overlap with either original or newly generated labels. Annotation names, abbreviations, parent structures, associated colors, previous 32-bit IDs, and current 16-bit IDs are organized in the DevCCF ontology (Supplementary Data 2), where red text indicates DevCCF modifications or additions to the ADMBA ontology. Warped CCFv3 segmentation values outside the 16-bit range were also translated to 16-bit values for ease of use. Translated values can be viewed

with original values in our modified CCFv3 ontology by viewing side-by-side columns 'id' (original values), and 'id_16bit' (new values) in the data download.

### GABAergic neuron quantification

We used LSFM to image *Gad2*-Cre;Ai14 mouse embryos at E11.5 (3 male, 4 female), E13.5 (1 male, 3 female, 1 unknown), and E15.5 (1 male, 2 female). We used the interactive machine learning for (bio)image analysis (ilastik) v1.3.3post3[103] to train a pixel classification-based machine learning model for each age to identify GAD2 positive (GAD2 + ) voxels in the whole embryo at $1.8 \times 1.8 \times 5 \mu m^3$ voxel resolution. We resampled the pixel classification images to 20 μm isotropic resolution for image registration to the DevCCF, where each voxel value represented the sum of GAD2+ voxels in the respective full resolution image. The GAD2+ voxel count was deformably registered to age matched DevCCF templates using ANTs call 'antsRegistrationSyN.sh' as described in multimodal registration methods. GAD2+ voxel count relative occupancy per anatomical region was calculated as a ratio of the sum of positive voxels to the number of voxels per region.

### Interactive web visualization development

The DevCCF web app, built using Neuroglancer v3.5, provides an interactive way to explore the developing mouse brain atlas. A key feature is the "Ontology" button, which allows users to navigate the hierarchical structure of brain regions defined by the DevCCF ontology. This functionality is achieved through a combination of HTML, JavaScript, and the Neuroglancer API. First, the web app loads the necessary libraries and data, including the atlas template, annotations, and a JSON file representing the initial viewer state. A custom script then parses a CSV file containing the DevCCF ontology structure, which defines each brain region with its ID, name, acronym, parent region, and color. From this parsed data, the script builds a hierarchical JSON structure representing the parent-child relationships between brain regions. Next, the script creates HTML elements for each brain region, assigns unique IDs, and sets the text content and color based on the CSV data. These elements are then arranged hierarchically, with child regions nested within their parent regions. Initially, only the top-level regions are visible. The "Ontology" button is added to the interface, and when clicked, it reveals the ontology container with the hierarchical tree of brain regions. Users can expand or collapse parent regions using "+" and "-" icons and click on individual regions to highlight them in the Neuroglancer viewer. The script leverages the Neuroglancer API to control the visibility of brain regions based on user interactions with the ontology tree. Selecting a region in the tree updates the corresponding segmentation layer's segment property in the Neuroglancer state, ensuring only the selected region and its descendants are visible. Conversely, changing the visible segments in the viewer updates the checked state of the corresponding checkboxes in the ontology tree, maintaining consistency between the two. Additional features include a search bar within the ontology container for finding specific regions and a hover function that displays detailed information about each region. This integration of HTML, JavaScript, and the Neuroglancer API provides a user-friendly and interactive way to explore the DevCCF ontology and visualize the corresponding brain regions.

### Aligning postnatal atlases to stereotaxic coordinates

Postnatal atlases were rotated to stereotaxic coordinates by aligning the estimated bregma and the anterior commissure in a common coronal plane. For each postnatal age, the MRI FA and T2w templates were opened in ITK-SNAP. Using 2D cross sections of the FA template and the 3D volume rendering of the T2w template, we annotated two midline points: the estimated bregma above the skull impression on the superior surface of the brain, and the center of the anterior commissure. Templates were rotated in the sagittal plane to place the estimated bregma directly superior to the anterior commissure.

To display the coordinate system, we created 3D grid images for each postnatal atlas at native MRI resolution. The grids can be overlayed on the age-matching atlas. The central vertex is over the anterior commissure. Space between gridlines represents 1 mm. Datasets are coronally oriented. Grid labels parallel to the sagittal, horizontal, and coronal planes are labelled 1, 2, and 3 respectively, allowing differentiation when overlayed. Numeric spacing labels were created using FIJI text tools and given values of 4. These labels are visible from the coronal plane, displaying height and width in millimeters, with anterior-posterior slice distance from the anterior commissure on bottom left corner of each section. To avoid confusion across various 3D visualization and analysis software, image metadata origins are fixed at (0, 0, 0).

### Volumetric growth curves

We calculated template average volume and standard deviation based on individual samples used to create each template. MRI DWI templates and annotations were aligned to each sample with ANTs 'antsRegistrationSyNQuick.sh.' Sample volumes were calculated for whole brain and selected parent regions through development: pallium, subpallium, terminal hypothalamus, peduncular hypothalamus, diencephalon, midbrain, prepontine hindbrain, pontine hindbrain, pontomedullary hindbrain, and medullary hindbrain. Volume averages and standard deviations were calculated using NumPy v1.21.1 and Pandas v2.2.0 libraries in python. Data was plotted with Prism v10.2.0 (GraphPad).

### Reporting summary

Further information on research design is available in the Nature Portfolio Reporting Summary linked to this article.

### Data availability

The DevCCF [https://doi.org/10.6084/m9.figshare.26377171][104] and unwarped LSFM templates [https://doi.org/10.6084/m9.figshare.26492122][105] generated in this manuscript have been deposited in FigShare. The DevCCF has been assigned RRID:SCR_025544. The DevCCF interactive viewer is available via the DevCCF hub [https://kimlab.io/brain-map/DevCCF/]. Individual sample MRI and LSFM datasets used to generate templates have been deposited in the Brain Imaging Library, accessible by searching for grant number RF1-MH124605 (https://api.brainimagelibrary.org/web/). CCFv3 and ADMBA datasets are available via the Allen Brain Atlas API. The Molecular Atlas is available at https://molecularatlas.org/. Supplementary Data files are provided with the supplementary data file. The Developmental Common Coordinate Framework (DevCCF) is an openly accessible resource via a Creative Commons Attribution 4.0 International License. Template, reference, annotation, and ontology version information are summarized in Supplementary Table 1. Additional information is available by contacting the corresponding author, Yongsoo Kim (yuk17@psu.edu). Source data are provided with this paper.

### Code availability

Custom in house code was generated for DevCCF development and data analysis. Code for template generation, multimodal registration, LSFM stitching, and 2D ISH data to the DevCCF is available via https://doi.org/10.5281/zenodo.12853683[106]. Code for mapping and quantifying annotations and cell type distribution between CCFv3 and P56 DevCCF is available at https://doi.org/10.5281/zenodo.12854081[107].

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

## Acknowledgements

We express gratitude to Yongsoo Kim Lab members and other DevCCF team members for their commitment, expertise, and motivation. We acknowledge the invaluable support of The Penn State College of Medicine High Performance Computing cluster. Our thanks to BioRender.com to generate Fig. 1a,b (under CC-BY license). We are grateful to the members of the BRAIN Initiative Cell Census Network for their insights. This work was supported by National Institutes of Health grants RF1MH12460501 (Y.K.), R01NS108407 (Y.K.), R01MH116176 (Y.K.), and R01EB031722 (J.G.). The contents are solely the responsibility of the authors and do not necessarily represent the views of the funding agency.

## Author contributions

Y.K. and L.N. conceptualized the project; R.B., J.K.L., J.A.M., D.S., S.B.M., J.X., Y.L. acquired LSFM data; J.Z. and C.H.L. acquired MRI data; Y.T.W. developed stitching pipeline; F.N.K., N.T., and J.G. performed template construction and image registration; F.N.K., R.P., and L.P. delineated anatomical segmentations; J.K.L. trained GABAergic machine learning model; A.B., and L.N. performed MERFISH Integration; J.T.D. and L.N. imported Allen developmental gene expression database; D.J.V., K.C.C. designed web visualization; Y.K., L.P., J.G., J.Z., and L.N. provided supervision; F.N.K. and Y.K. prepared the manuscript; All authors discussed and commented on the manuscript.

## Competing interests

The authors declare no competing interest.
