## [Peer Review File · Nature Communications]

Developmental Mouse Brain Common Coordinate FrameworkREVIEWER COMMENTS

Reviewer #1 (Remarks to the Author):

In this manuscript, the author introduces a novel multimodal 3D developmental common coordinate framework (DevCCF). This framework comprises a collection of 3D atlases, integrating MRI and LSFM templates, along with associated 3D annotations grounded in a developmental ontology. The application of DevCCF is demonstrated through various use cases, providing compelling evidence of its efficacy in facilitating robust data analysis.

The author presents a pipeline for the creation of seven DevCCF atlases, spanning from sample collection and MRI/LSFM imaging to 3D reconstruction and the construction of symmetric templates. The credibility of these atlases is underpinned by scholarly evidence. The DevCCF encapsulates undistorted morphology and intensity-averaged symmetric templates at pivotal developmental ages (E11.5, E13.5, E15.5, E18.5, P4, P14, and P56), aligning with the ADMBA and encompassing critical stages of regional differentiation.

Acknowledging the inherent variability among individual brains, the morphologically averaged templates, derived from both male and female samples, establish a shared reference morphology at each developmental age. At each stage, the DevCCF includes a minimum of four MRI contrast templates alongside an aligned, age-matched LSFM autofluorescence template. Furthermore, the establishment of DevCCF leverages the advantages of both undistorted MRI morphology and the cellular resolution of LSFM within the same spatial framework. This is achieved through the optimization of 3D landmark-assisted multimodal registration methods, facilitating the mapping of LSFM templates onto age-matched MRI templates. Consequently, this approach yields DevCCF templates that integrate information from multiple modalities.

Additionally, the author introduces developmentally consistent annotations rooted in the prosomeric model of mammalian brain anatomy, enhancing the utility and reliability of the DevCCF for data analysis.

The significance of DevCCF is underscored by its capacity to facilitate meaningful data analysis, a characteristic further demonstrated through various use cases. One notable application involves the exploration of new avenues for mapping and quantifying distinct cell types in developing brains, mirroring methodologies employed in adult mouse brains. Leveraging tissue clearing techniques and LSFM imaging at critical developmental stages (E11.5, E13.5, and E15.5), the authors successfully unveil the emergence of GABAergic neurons in embryonic brains.

Furthermore, the P56 DevCCF presents an opportunity to develop ontology-based labels with widely used CCFv3 annotations grounded in cytoarchitectures. This comparative analysis not only highlights the robustness of the DevCCF but also establishes its compatibility with established annotation frameworks. The outcomes of these data analyses carry biological significance, contributing insights to our understanding of developmental processes.

This paper's pipeline shows the robustness of the 3D developmental mouse brain atlases, along with their validity. The DevCCF emerges as a tool facilitating the interpretation and integration of diverse data across modalities and scales, including advanced high-resolution 3D imaging and spatial genome data. It is demonstrated through practical use cases, such as the examination of GABAergic neuron spatiotemporal development and CCFv3-aligned data analysis within a developmental framework.

The DevCCF, theoretically, opens a door to broader practical applications. The author successfully aligns the CCFv3 template and annotations with DevCCF postnatal templates, enabling the seamless integration of datasets previously registered to the CCFv3 with the DevCCF. This integration paves the way for large-scale data analysis, accommodating the cell census data.

However, the DevCCF acknowledges room for improvement, particularly in the need for more extensive sample data for each age and the inclusion of samples from additional modalities. The integration of artificial intelligence algorithms in template construction is also identified as a potential enhancement. For instance, ensuring the accuracy of cross-age registration and interpolation can substantiate the consistency of brain developmental stages across the atlases.

Some suggestions:

1/ In the Introduction section, the author can enhance the literature review on developmental 3D reference atlases to provide a more comprehensive context for the research question. Also it is important to discuss use of the recent atlas-construction tools such as mBrainAligner to map MRI data with the atlas templates, as showcased in recent papers such as Han et al, <https://www.science.org/doi/10.1126/sciadv.adf3771> .

2/ In the Method section, the author should offer additional details and present their research methodology in a more vivid manner. This approach will aid readers in comprehending the entire pipeline more easily.

3/ In the Discussion section, the author has the opportunity to provide a concise analysis of the listed limitations of DevCCF and discuss the feasibility of addressing these limitations in future efforts. Also maybe with the enhancement of various registration tools.

4/ In the Results section, the author can improve the differentiation of the subtitles 'DevCCF allows examination of GABAergic neuron spatiotemporal development' and 'DevCCF enables CCFv3 aligned data analysis with a developmental framework' to make them stand out more prominently from the rest of the content.

Reviewer #2 (Remarks to the Author):

Kronman et al., present a novel developmental atlas framework for the embryonic and early postnatal mouse brain based on co-registered MRI and lightsheet image volumes, comprising seven atlases for different stages of the developing mouse brain. The atlases are provided with anatomical segmentations following a prosomeric developmental schema. The developmental atlas delineations are related to the adult mouse brain CCF3 atlas via the reference image data for the P56 brain. The use of the atlas resources is exemplified by mapping the development of GABAergic neurons.

The presented series of developmental atlases stand out from currently available atlases by being defined in high quality undistorted volumetric image data (in skull ex vivo MRI) with additional detail from co-registered detailed lightsheet microscopic data, and by having detailed delineations tailored for the reference images at the developmental stages included. As such, the developmental common coordinate framework presented has a good potential for becoming an impactful and valuable resource for the neuroscience community. The paper is well written, the methods sound, and the Figures are nicely presented.

The paper presents a novel atlas resource that merits publication and likely will have positive impact for the field. There are, however, several important questions and issues that should be resolved:

1. The atlas data are shared via the web page of the Kim laboratory as an openly accessible resource, with images downloadable from a private Sharepoint drive. But technical descriptions of the shared files and explanation of file organization is missing. The shared data lack versioning, license, information about conditions for use, and citation information. To be Findable, Accessible,

Interoperable, and Reuseable, the data should be shared in a public repository with clear license, a DOI and RRID assigned.

2. All image volumes spatially integrated in MRI space are shared, but given the applied deformations, some volumes appear distorted and at some levels difficult to interpret. It would be useful if the non-deformed average image volumes also were shared, preferably including individual volumes.

3. In the Allen CCF version deformed to DevCCF space (P56_CCFv3_20mm.nii.gz) 21 labels lack names. Some of these correspond to entire regions, others are only a few voxels, some include voxels from several regions. This is problematic and must be corrected. This problem may also have implications for the analysis in Figure 5 that should be considered.

4. The mapping of the adult mouse CCFv3 delineations onto the prosomeric delineation schema in the P56 developmental atlases is valuable, but while this provides opportunity for direct comparison of delineations at the P56 stage, such comparisons will be difficult to translate to atlases for younger brains, unless a similar warping is produced for all seven atlases. It is not likely that the CCFv3 delineations will have similar relations to the prosomeric delineations at earlier stages. As currently presented it will remain difficult to relate findings from prenatal and early postnatal mouse brains to adult neuroarchitecture.

5. The atlas reference images were made symmetric and averaged between sexes, not necessarily a trivial or obvious choice, given the available literature on asymmetries and sex differences in rodent brains. The motivation for create a unilateral atlas should therefore be introduced and possible consequences or limitations should be discussed.

6. While the population averaged images volumes are symmetric, the annotations for DevCCF are only delineated and shown in one hemisphere. This substantially reduces the utility of the atlas. It should at least be described how users can use the atlas for bilateral analyses, with discussion of possible caveats or limitations. It further is confusing that the P4 template shown in the viewer tool is delineated on the opposite side compared to the other atlases, this should be corrected or at least explained. The uptake of the atlas would likely increase if whole-brain, bilateral delineation volumes were provided.

7. The atlas delineations are described to adhere to prosomeric principles derived from the genoarchitecturally defined delineations of the 2D ADMBA and validated using spatially aligned in-situ hybridization (ISH) data, supplied with information from multiple other image modalities. It would be very useful to see example documentation of detailed correspondence between different

imaging modalities and ISH data, e.g. in supplementary figures. Such figures could show how the different MRI modalities were used to identify boundaries. Here it would also be useful to see demonstrated how the (10 micrometer resolution) LSM images were utilized.

8. Reference atlases serve many purposes, both as an anatomical reference, a common space for data integration, a region of interest standard for analyses, and a general aid for communication about brain structure and organization. When atlases are used to define and communicate the location of observations, it is very important that the criteria underlying delineations are transparent and explicitly defined. This requires documentation of criteria and strict versioning of the reference data, delineations, and nomenclature. The same goes for the (stereotaxic?) coordinates provided in the viewer tool. How was the coordinate system applied, with origin, orientation, and scaling? How were skull landmarks identified, how do stereotaxic coordinates relate to the native image space? All such technical parameters need to be documented and versioned. Awareness about this is increasing in the field and recommendations can be found in the literature. Sufficient technical documentation should be provided with the data to allow users to use the different elements of the atlases in analyses, tools, and communication about anatomical locations. This should include unique naming and IDs, versioning, and definitions underlying delineations and coordinates used to navigate atlases.

9. Although white matter was used as a starting point for delineations, several important white matter (landmark) regions are not delineated in DevCCF, e.g. the anterior commissure. It is understandable that certain regions may be difficult to identify, but it should be explained in the text with more detail.

10. Some of the methods descriptions are too succinct to be fully reproducible, e.g. the procedure describing 2D in situ hybridization gene expression mapping onto the DevCCF, what procedure, software or code was used, are all tools and codes openly shared?

11. The label files provided only contain abbreviations. Full names are given in the Excel files, but it would be more user friendly to include full names in the label file, so that all information is consistently presented in one file. The (red) colour code used in the spreadsheets is not explained.

Reviewer #3 (Remarks to the Author):

In their manuscript, Kronman et al, have developed a common coordinate framework (DevCCF) for the developing mouse brain, including early postnatal development (P4 and P14). They generated the DevCCF using MRI and co-registration with images from light sheet microscopy. This effort is important and can help future comparative studies that focus on development of brain regions or

migration of cell types. A study from the Allen Institute established a 3D atlas (Thompson et al, 2014) with improvements in implementation (Young et al, 2021) , but the authors claim that their DevCCF atlas is superior to these efforts - this point is not convincingly demonstrated nor is it clearly explained or defined. The authors used the same nomenclature and organization as the Allen Institute atlas, which should facilitate side by side comparisons and help the community explore both atlases. The authors also present a simple interface for the web-based visualization of the DevCCF atlas, which further can help disseminate their work.

The authors used 7 developmental ages for the DevCCF, but it was unclear from the main text how many samples that were processed per stage and the corresponding variance (a supplementary table shows number of samples). In comparison the CCFv3 reference atlas and the Allen developmental atlas was produced based on a very large sample size (hundreds of brains imaged at high resolution). The authors should provide some quantification and visualization of the averaged atlas and how variance impacts on the segmentation of different brain regions, including the possible effects between males and females. The authors should be more explicit in their text on the advantages and disadvantages of their approach and how their data quality compares to published atlases.

The authors used the DevCCF to map the migration of GABAergic cells (Figure 4), but it is unclear how this mapping reveals anything new on the GABAergic cell migration and the authors do not interpret their findings in light of the literature.

The authors show in Figure 5 a comparison between DevCCF and CCFv3: this is an important comparison but this aspect of the work is not fully explored in the manuscript. Here the authors could do a more ambitious comparison between the two atlases to clearly show where they agree or disagree, and present some evidence for example using the ISH data on the reasons from discrepancies between atlases.

The authors performed an interesting analysis and compared the DevCCF with the CCFv3 reference atlas, focusing on mapping cell type classification using a MERFISH dataset – this comparison did not directly address how cell type composition can be used to refine or adjust segmentation or definition of brain regions. Again, the analysis seems a bit superficial and lacks any reflection on findings or how to interpret them.

To directly address the issue of how different biological signals can be used to define brain subregions, the authors should compare their DevCCF with the molecular atlas based on spatial transcriptomics (Ortiz et al, 2020), which defined cortical and subcortical regions based on the gene expression patterns in the adult mouse brain – it would be of great interest to compare the developmentally-defined regions in DevCCF in the adult (P56) with the regions defined by spatial transcriptomics to conclude how these two different definitions converge or diverge. The authors raised this important issue in their Discussion: to what extent different atlases agree on definitions

and segmentation of subregions, and should take advantage of their DevCCF to provide some new insight into this question.

REVIEWER COMMENTS

Reviewer #1 (Remarks to the Author):

In this manuscript, the author introduces a novel multimodal 3D developmental common coordinate framework (DevCCF). This framework comprises a collection of 3D atlases, integrating MRI and LSFM templates, along with associated 3D annotations grounded in a developmental ontology. The application of DevCCF is demonstrated through various use cases, providing compelling evidence of its efficacy in facilitating robust data analysis.

The author presents a pipeline for the creation of seven DevCCF atlases, spanning from sample collection and MRI/LSFM imaging to 3D reconstruction and the construction of symmetric templates. The credibility of these atlases is underpinned by scholarly evidence. The DevCCF encapsulates undistorted morphology and intensity-averaged symmetric templates at pivotal developmental ages (E11.5, E13.5, E15.5, E18.5, P4, P14, and P56), aligning with the ADMBA and encompassing critical stages of regional differentiation.

Acknowledging the inherent variability among individual brains, the morphologically averaged templates, derived from both male and female samples, establish a shared reference morphology at each developmental age. At each stage, the DevCCF includes a minimum of four MRI contrast templates alongside an aligned, age-matched LSFM autofluorescence template. Furthermore, the establishment of DevCCF leverages the advantages of both undistorted MRI morphology and the cellular resolution of LSFM within the same spatial framework. This is achieved through the optimization of 3D landmark-assisted multimodal registration methods, facilitating the mapping of LSFM templates onto age-matched MRI templates. Consequently, this approach yields DevCCF templates that integrate information from multiple modalities.

Additionally, the author introduces developmentally consistent annotations rooted in the prosomeric model of mammalian brain anatomy, enhancing the utility and reliability of the DevCCF for data analysis.

The significance of DevCCF is underscored by its capacity to facilitate meaningful data analysis, a characteristic further demonstrated through various use cases. One notable application involves the exploration of new avenues for mapping and quantifying distinct cell types in developing brains, mirroring methodologies employed in adult mouse brains. Leveraging tissue clearing techniques and LSFM imaging at critical developmental stages (E11.5, E13.5, and E15.5), the authors successfully unveil the emergence of GABAergic neurons in embryonic brains.

Furthermore, the P56 DevCCF presents an opportunity to develop ontology-based labels with widely used CCFv3 annotations grounded in cytoarchitectures. This comparative analysis not only highlights the robustness of the DevCCF but also establishes its compatibility with established annotation frameworks. The outcomes of these data analyses carry biological significance, contributing insights to our understanding of developmental processes.

This paper's pipeline shows the robustness of the 3D developmental mouse brain atlases, along with their validity. The DevCCF emerges as a tool facilitating the interpretation and integration of diverse data across modalities and scales, including advanced high-resolution 3D imaging and spatial genome data. It is demonstrated through practical use cases, such as the examination of GABAergic neuron spatiotemporal development and CCFv3-aligned data analysis within a developmental framework.

The DevCCF, theoretically, opens a door to broader practical applications. The author successfully aligns the CCFv3 template and annotations with DevCCF postnatal templates, enabling the seamless integration of datasets previously registered to the CCFv3 with the DevCCF. This integration paves the way for large-scale data analysis, accommodating the cell census data.

We appreciate the detailed summary and enthusiasm on our manuscript.

However, the DevCCF acknowledges room for improvement, particularly in the need for more extensive sample data for each age and the inclusion of samples from additional modalities. The integration of artificial intelligence algorithms in template construction is also identified as a potential enhancement. For instance, ensuring the accuracy of cross-age registration and interpolation can substantiate the consistency of brain developmental stages across the atlases.

We appreciate the constructive critique. We provide a detailed response below.

Some suggestions:

1. In the Introduction section, the author can enhance the literature review on developmental 3D reference atlases to provide a more comprehensive context for the research question. Also it is important to discuss use of the recent atlas-construction tools such as mBrainAligner to map MRI data with the atlas templates, as showcased in recent papers such as Han et al, <https://www.science.org/doi/10.1126/sciadv.adf3771> .

To enhance the literature review on developmental 3D reference atlases, we have detailed the needs of a developmental atlas, shared more features and limitations about the previous developmental atlas (ADMBA), and expanded upon the use and limitations of MRI-based developmental mouse brain atlases, and added a table comparing recent developmental atlases in the introduction. The Han et al. paper linked does not refer to mBrainAligner. However, we have noted that recent brain mapping tools, such as mBrainAligner enable cross modal mapping of 3D datasets to the CCFv3, and citing alternative sources.

Following texts have been added in the introduction.

“Recent brain mapping tools, such as mBrainAligner^{22,23} and Multimodal 3D Mouse Brain Atlas Framework with the Skull-Derived Coordinate System²⁴, have enabled cross modal mapping of MRI, LSFM, and other 3D datasets to the CCFv3, advancing the use of this mouse atlas^{6,25}.”

“Therefore, developmental research requires a standard reference atlas to comprise several templates and annotations that span multiple stages of development and remain connected by a common anatomical schema. Several developmental atlases have been published varying by imaging modality, accessibility, age ranges, labeled detail, and template specifications (Table 1).”

2. In the Method section, the author should offer additional details and present their research methodology in a more vivid manner. This approach will aid readers in comprehending the entire pipeline more easily.

Thank you for this note. In order to help readers understand our methods, we have added additional details about our imaging techniques, symmetric template construction, multimodal registration, gene expression mapping to the DevCCF, Differentiating CCFv3 and Adults DevCCF data, anatomical segmentations, GABAergic neuron quantification, our web platform development, stereotaxic coordinates, and volumetric analysis.

3. In the Discussion section, the author has the opportunity to provide a concise analysis of the listed limitations of DevCCF and discuss the feasibility of addressing these limitations in future efforts. Also maybe with the enhancement of various registration tools.

We have added to the discussion details about the current limitation and potential future improvement on DevCCF templates and annotations, as well as how registration tools can aid in improving template building, multimodal registration, and user registration.

Following texts have been added in the discussion.

“The DevCCF is not without limitations. Increasing sample size and enhancing registration methods could improve template signal contrast, enabling updates to multimodal alignment and anatomical parcellation, and improving experimental data alignment^{15,54}. Increasing sample size will be possible as continue to share 3D whole brain data via open repositories, such as OpenNeuro and Brain Imaging Library⁵⁵. Registration enhancement is possible with advancement of registration tools such as Elastix⁷³, FSL⁷⁴, and ANTs³⁸. This will also improve multimodal registration,^{23,24,53,54} and inclusion of additional 3D imaging modalities such as micro-computed tomography, which could expand the DevCCF's utility^{23,24,75}. On the user end, expansion of multimodal atlas registration software such as mBrainAligner^{22,23} and BrainGlobe⁷⁶ to include the DevCCF would simplify experimental data integration. Future refinement of anatomical segmentations is warranted including white matter tracks. We also expect that experimental discoveries and automated segmentation based on diverse mapped data will necessitate further segmentation and updates to DevCCF annotations^{77,78}.

4. In the Results section, the author can improve the differentiation of the subtitles 'DevCCF allows examination of GABAergic neuron spatiotemporal development' and 'DevCCF enables CCFv3 aligned data analysis with a developmental framework' to make them stand out more prominently from the rest of the content.

These section titles have been modified to: 'Charting early emergence of GABAergic neurons using DevCCF' (from 'DevCCF allows examination of GABAergic neuron spatiotemporal development'), and 'Integration of CCFv3 with P56 DevCCF for data analysis with a developmental framework' (from 'DevCCF enables CCFv3 aligned data analysis with a developmental framework')

Reviewer #2 (Remarks to the Author):

Kronman et al., present a novel developmental atlas framework for the embryonic and early postnatal mouse brain based on co-registered MRI and lightsheet image volumes, comprising seven atlases for different stages of the developing mouse brain. The atlases are provided with anatomical segmentations following a prosomeric developmental schema. The developmental atlas delineations are related to the adult mouse brain CCF3 atlas via the reference image data for the P56 brain. The use of the atlas resources is exemplified by mapping the development of GABAergic neurons.

The presented series of developmental atlases stand out from currently available atlases by being defined in high quality undistorted volumetric image data (in skull ex vivo MRI) with additional detail from co-registered detailed lightsheet microscopic data, and by having detailed delineations tailored for the reference images at the developmental stages included. As such, the developmental common coordinate framework presented has a good potential for becoming an impactful and valuable resource for the neuroscience community. The paper is well written, the methods sound, and the Figures are nicely presented.

The paper presents a novel atlas resource that merits publication and likely will have positive impact for the field. There are, however, several important questions and issues that should be resolved:

We appreciate the positive evaluation of our work. Detailed responses to the constructive critiques are below.

1. The atlas data are shared via the web page of the Kim laboratory as an openly accessible resource, with images downloadable from a private Sharepoint drive. But technical descriptions of the shared files and explanation of file organization is missing. The shared data lack versioning, license, information about conditions for use, and citation information. To be Findable, Accessible, Interoperable, and Reuseable, the data should be shared in a public repository with clear license, a DOI and RRID assigned.

We added "READ_ME.txt" file in our download folder that contains technical details. We also added Supplementary Table 6 for versioning. Datasets will be deposited in the FigShare public repository with a unique DOI upon publication. Once deposited, we will request the RRID via scicrunch.org/resources. The DevCCF is available with CC BY 4.0 (<https://creativecommons.org/licenses/by/4.0/>) license.

2. All image volumes spatially integrated in MRI space are shared, but given the applied deformations, some volumes appear distorted and at some levels difficult to interpret. It would be useful if the non-deformed average image volumes also were shared, preferably including individual volumes.

We have made the unwarped average LSFM templates available for download with the DevCCF in the public repository. Unwarped individual datasets are available for download via the Brain Imaging Library

(<https://api.brainimagelibrary.org/web/list?fields=Grant&searchtxt=RF1-MH124605>). Datasets can be found by searching subject numbers, also made available via Supplementary Table 1. We have added a separate 'Data Availability' section to the manuscript to detail this download information.

3. In the Allen CCF version deformed to DevCCF space (P56_CCFv3_20mm.nii.gz) 21 labels lack names. Some of these correspond to entire regions, others are only a few voxels, some include voxels from several regions. This is problematic and must be corrected. This problem may also have implications for the analysis in Figure 5 that should be considered.

We have checked the warped Allen CCFv3 annotation data in the Neuroglancer visualizer and via download and viewing in Fiji. We found that all values present in this annotation set are CCFv3 annotations according to the Allen CCFv3 ontology. CCFv3 annotations are 32-bit images, which exceeds the 16-bit range of many data viewers, including the current release of ITK-SNAP (but not in alpha release). These viewers may clip or artificially generate alternate IDs for 32-bit images. To avoid this issue in the future, we have generated a 16-bit version of CCFv3 annotations in our data share location, where all values outside the 16-bit range are translated to values in the 16-bit range. The translation can be viewed side-by-side with original values in our modified CCFv3 ontology, by viewing the columns labelled id (original values), and id_16bit (new values). This problem has not affected analysis comparing CCFv3 and DevCCF annotations, which was performed using python libraries Plotly and Seaborn, which do not clip 32-bit data.

4. The mapping of the adult mouse CCFv3 delineations onto the prosomeric delineation schema in the P56 developmental atlases is valuable, but while this provides opportunity for direct comparison of delineations at the P56 stage, such comparisons will be difficult to translate to atlases for younger brains, unless a similar warping is produced for all seven atlases. It is not likely that the CCFv3 delineations will have similar relations to the prosomeric delineations at earlier stages. As currently presented it will remain difficult to relate findings from prenatal and early postnatal mouse brains to adult neuroarchitecture.

Thank you for this feedback. We agree that it would be challenging to translate CCFv3 delineations to younger templates. CCFv3 delineations are specific to the adult mouse brain cytoarchitecture, and do not follow the prosomeric developmental schema. While we do not intend to produce warped CCFv3 labels for all ages, we have produced them for postnatal ages by serially warping them from CCFv3 to P56 DevCCF, then to P14 DevCCF, followed by P4 DevCCF. We see CCFv3 delineations can be aligned to younger ages, however, they may not be fully appropriate. For example, delineated cerebellar lobules are not fully developed by P4. Such issue at developmental stages reflects the need for the DevCCF.

To clarify this, we have specified that this integration is specific to P56 DevCCF, throughout the Results and Discussion section such as "Although distinct cytoarchitecture (e.g., Nissl visualization) have been useful to segment adult brain atlases^{12,13}, it is difficult to apply in developing brains due to immature and rapidly migrating cells."

To further aid the comparison between CCFv3 and P56 DevCCF, we created Supplementary Table 4 to compare voxel-to-voxel correspondence between the two atlas labels.

5. The atlas reference images were made symmetric and averaged between sexes, not necessarily a trivial or obvious choice, given the available literature on asymmetries and sex differences in rodent brains. The motivation for create a unilateral atlas should therefore be introduced and possible consequences or limitations should be discussed.

While our templates are symmetric, they are not unilateral. Rather they are unbiased morphological averages of combined left and right hemispheres. Previous atlases have been generated either symmetric (e.g., CCFv3), or provided annotations for only a single hemisphere, providing convention that morphology does not differ enough between hemispheres to warrant a lateralized atlas. Asymmetries and sex differences are both small in a morphological sense and require large sample sizes to accurately detect¹. Therefore, they are accounted for during template alignment to user data. Template symmetry accommodates data with unknown laterality, as data is often published without regard to laterality or with data from multiple hemispheres combined. Atlas symmetry does not preclude use with lateralized data, but eliminates the potential for user error aligning lateralized data, and removes the confusion of which hemisphere to select with unknown laterality. An atlas with the goal to integrate diverse datasets must allow for ambiguous laterality. Similarly, averaging sexes does not preclude use with males or females, rather it provides an unbiased starting point that may be used to use asses potential sexual dimorphism by registering symmetrical data to samples from both sexes.

We added the following text in the result section.

“Moreover, symmetrical templates can serve as initial space to assess potential laterality and sexual dimorphism of the brain by registering to individual samples.”

6. While the population averaged images volumes are symmetric, the annotations for DevCCF are only delineated and shown in one hemisphere. The substantially reduces the utility of the atlas. It should at least be described how users can use the atlas for bilateral analyses, with discussion of possible caveats or limitations. It further is confusing that the P4 template shown in the viewer tool is delineated on the opposite side compared to the other atlases, this should be corrected or at least explained. The uptake of the atlas would likely increase if whole-brain, bilateral delineation volumes were provided.

We agree that bilateral delineation will improve atlas usefulness. Since our templates are symmetrical, we made bilateral annotations for the whole template to cover both hemispheres. We have kept viewer annotations half brain for easier side-by-side viewing and corrected the P4 annotations that were flipped on the opposite side in the web viewer.

7. The atlas delineations are described to adhere to prosomeric principles derived from the genoarchitectonically defined delineations of the 2D ADMBA and validated using spatially aligned in-situ hybridization (ISH) data, supplied with information from multiple other image modalities. It would be very useful to see example documentation of detailed correspondence between different imaging modalities and ISH data, e.g. in supplementary figures. Such figures

could show how the different MRI modalities were used to identify boundaries. Here it would also be useful to see demonstrated how the (10 micrometer resolution) LSFM images were utilized.

We added a new Figure 3, Supplementary Fig 4 and table 3, documenting how ISH data was used for validation alongside MRI and LSFM imaging data. This is described in Results section '3D Anatomical labels with an updated developmental ontology' and related method section such as the following text.

"We used landmarks visible in DevCCF templates, as well as aligned 3D imaging and side-by-side reference materials as evidence to draw 3D anatomical segmentations (Fig. 3, Table 3, Supplementary Fig. 4). Several template contrasts help delineate the brain surface-to-surface areas at the choroidal tissue, pallium, and brainstem (Fig. 3a,g,m) and cerebellum and midbrain (Fig. 3m), as well as surface to skull areas (Fig. 3n-r). Further segmentation was continued at increasing ontological depth, to at least Level 5 (Fig. 4a).

The DevCCF segmentations are simplest at E11.5 and grow in complexity and number through P56 (Fig. 4b,c; Supplementary Table 2). E11.5 annotations consist of ventricles, neuromeric boundaries in the rostral to caudal direction; floor, basal, alar, and roof boundaries in the ventral to dorsal direction; and pallium, subpallium divisions of the telencephalon (Fig. 4b,c). E13.5 annotations include additional subpallial segmentations as well as early stages of the concentric ring topology, defining early divisions of the neocortex, allocortex, and mesocortex (Fig. 4b,c). Cortical layers are segmented as early as P4, and cerebellar layers are segmented as early as P14 (Fig. 4b,c).

Segmentation was aided and validated by 3D histological staining from LSFM with alignment to the DevCCF (Fig. 4d-f). For example, neurofilament staining supports neocortical layer delineation, with the strongest staining in layer 4, and distinguishes mesocortical olfactory, insular, hippocampal, and cingulate cortices (Fig. 4e). Aligned fluorescence Nissl images similarly aided P56 cerebellar internal granular layer, and olfactory bulb layers delineation (Fig. 4f). Segmentations are mutually exclusive, meaning all brain voxels are labeled uniquely with a single ontology ID (Supplementary Table 2).

Lastly, we imported 2D sectioned ADMBA in situ hybridization (ISH) gene expression data onto age matched DevCCF templates to validate our anatomical segmentation (Supplementary Fig. 4). We confirmed that gene expression profiles registered to the DevCCF are concordant with our template signal and anatomical segmentations at each developmental age (Supplementary Fig. 4)"

We would like to clarify that unwarped LSFM templates are 10 micron isotropic, while warped templates are limited to 20 micron isotropic resolution for computational efficiency during multimodal registration and to keep data size more manageable for most personal computers. This has been reflected in the manuscript methods ('Multimodal registration of 3D imaging to the DevCCF').

8. Reference atlases serve many purposes, both as an anatomical reference, a common space for data integration, a region of interest standard for analyses, and a general aid for communication about brain structure and organization. When atlases are used to define and communicate the location of observations, it is very important that the criteria underlying delineations are transparent and explicitly defined. This requires documentation of criteria and strict versioning of the reference data, delineations, and nomenclature. The same goes for the (stereotaxic?)

coordinates provided in the viewer tool. How was the coordinate system applied, with origin, orientation, and scaling? How were skull landmarks identified, how do stereotaxic coordinates relate to the native image space? All such technical parameters need to be documented and versioned. Awareness about this is increasing in the field and recommendations can be found in the literature. Sufficient technical documentation should be provided with the data to allow users to use the different elements of the atlases in analyses, tools, and communication about anatomical locations. This should include unique naming and IDs, versioning, and definitions underlying delineations and coordinates used to navigate atlases.

Thank you for providing these detailed suggestions.

First, we have incorporated suggested information in the ReadMe file of the DevCCF download website, detailing the atlas organization.

Second, we have provided a new table (Supplementary Table 6) including unique data for versioning, reference data, delineations, ontology, coordinates, IDs, versioning, license, and unique names. We also added more details in our anatomical delineation in the method section: 'Anatomical segmentations for the DevCCF' such as the following text.

"Neuromeres and pallial cortex division boundaries follow smooth trajectories that end perpendicular to the brain surface and the ventricles. Each neuromere stretches from floor to roof plate and is in contact with only one neuromere caudally and one neuromere rostrally³². Pallial neocortex (NeoCx), allocortex (AlloCx), and mesocortex (MesoCx) are segmented according to the concentric ring topology as 6-layer structures, 3-layer structures, and transitional structures, respectively, where layers are overlaid from ventricular to superficial (Supplementary Fig. 3)³⁹. Choroidal tissue is within the roof plate. The medullary hindbrain (MH) neuromeres are divided evenly into 5 segments (r7-r11). Parcellations are based on previously defined landmarks^{28,32,39,42,97-99} primarily visualized with template contrast features. For example, the lateral and medial ganglionic eminences mark the early subpallium (Supplementary Fig. 4a). The facial motor nerve identifies the ventral surface and alar-basal boundary of rhombomere 6 (r6) (Supplementary Fig. 4e). In late postnatal ages, the neocortex, allocortex, and mesocortex were segmented by separating regions by thick cortical layer areas, 3-layer areas, and transitional zones, respectively, visible in MRI templates (Supplementary Fig. 4f,h). Regions without easily visible landmarks are delineated based on neighboring regions. Mesomere 2 (m2), for example is a thin neuromere between the caudal most landmarks of mesomere 1 (m1) and the rostral most landmarks of isthmus (r0) (Table 3)."

Third, we added supplementary Fig. 2 to show our approach to define stereotaxic coordinates and the resulting grid file for overlay. Methodological details for the stereotaxic coordinate have been added in the Method section: 'Aligning postnatal atlases to stereotaxic coordinate'. Descriptions of stereotaxic coordinates were refined in results and discussion to detail landmark identification. We included information about the origin, orientation, and scaling resolution. Briefly, bregma was estimated based on imprints on the brain. Anterior commissure was defined based on MRI fraction anisotropy. The image was rotated to align bregma and anterior commissure in the coronal plane. The templates were centered in the field of view. All templates are resliced to LSP orientation (coronal). We hope this information will help readers understand our protocol and help users understand the dataset for ease of use.

9. Although white matter was used as a starting point for delineations, several important white matter (landmark) regions are not delineated in DevCCF, e.g. the anterior commissure. It is understandable that certain regions may be difficult to identify, but it should be explained in the text with more detail.

We acknowledge that the detailed segmentation of white matter tracts is not included in our current version. These tracts were visualized by annotators, but not delineated. We added descriptions of how landmarks were used for DevCCF delineations in Results: 'Anatomical segmentations are defined by landmarks', including new Figure 3 and supplementary fig. 4 showing many landmarks for an E13.5 example. Moreover, we added Table 3, listing common landmarks to delineate neuromeres and alar plates. We also added the following sentence in the discussion, under the limitation of our DevCCF.

"Future refinement of anatomical segmentations is warranted including white matter tracks."

10. Some of the methods descriptions are too succinct to be fully reproducible, e.g. the procedure describing 2D in situ hybridization gene expression mapping onto the DevCCF, what procedure, software or code was used, are all tools and codes openly shared?

We enhanced the method section with further technical details. For example, to help readers understand the procedure describing 2D in situ hybridization gene expression mapping onto the DevCCF, we greatly expanded "2D in situ hybridization gene expression mapping onto the DevCCF" in the methods section and shared the related code publicly on GitHub, as newly noted in the Code Availability section.

11. The label files provided only contain abbreviations. Full names are given in the Excel files, but it would be more user friendly to include full names in the label file, so that all information is consistently presented in one file. The (red) colour code used in the spreadsheets is not explained.

We have updated the ITK-SNAP label description files for both DevCCF and CCFv3 to include both abbreviations and full name. We agree this suggestion should make the DevCCF more user friendly when exploring the atlas using ITK-SNAP. Additionally, we have added a description of the red color code in DevCCF ontology, indicating it indicates DevCCF modifications and updates to the Allen Developing Mouse Brain Atlas ontology.

Reviewer #3 (Remarks to the Author):

In their manuscript, Kronman et al, have developed a common coordinate framework (DevCCF) for the developing mouse brain, including early postnatal development (P4 and P14). They generated the DevCCF using MRI and co-registration with images from light sheet microscopy. This effort is important and can help future comparative studies that focus on development of brain regions or migration of cell types. A study from the Allen Institute established a 3D atlas (Thompson et al, 2014) with improvements in implementation (Young et al, 2021) , but the authors claim that their DevCCF atlas is superior to these efforts - this point is not convincingly demonstrated nor is it clearly explained or defined. The authors used the same nomenclature and organization as the Allen Institute atlas, which should facilitate side by side comparisons and help the community explore both atlases. The authors also present a simple interface for the web-based visualization of the DevCCF atlas, which further can help disseminate their work.

We appreciate overall enthusiasm on our manuscript. To set the stage of our manuscript, we updated our introduction with better organized text including the need to improve current developmental atlases. For example, we revised the following text.

“Yet, ADMBA is based on a single animal per age and offers limited usage to accommodate emerging high-resolution 3D whole brain data. Although recent 3D atlases interpolated and extrapolated from the ADMBA helped to generate 3D templates and annotations³³, single modality-based templates and smoothed annotations without biological validation require further improvement.”

1. The authors used 7 developmental ages for the DevCCF, but it was unclear from the main text how many samples that were processed per stage and the corresponding variance (a supplementary table shows number of samples). In comparison the CCFv3 reference atlas and the Allen developmental atlas was produced based on a very large sample size (hundreds of brains imaged at high resolution). The authors should provide some quantification and visualization of the averaged atlas and how variance impacts on the segmentation of different brain regions, including the possible effects between males and females. The authors should be more explicit in their text on the advantages and disadvantages of their approach and how their data quality compares to published atlases.

We added Table 2 to provide details on DevCCF sample size and variability of brain volume. We also created Supplementary Figure 5 and Supplementary Table 3 to provide detailed volume quantification of individual areas and their variance across samples and developmental time.

While DevCCF templates do not meet the sample size of the 1675 sample adult CCFv3, it surpasses the sample size of previous 3D developmental mouse brain templates, often limited to one sample per template as illustrated in Table 1. However, we also acknowledge to improve our template with more samples in the future. We added the following text in the limitation section of the discussion.

“Increasing sample size and enhancing registration methods could improve template signal contrast, enabling updates to multimodal alignment and anatomical parcellation, and improving experimental

data alignment^{15,54}. Increasing sample size will be possible as continue to share 3D whole brain data via open repositories, such as OpenNeuro and Brain Imaging Library⁵⁵.”

Moreover, we added the following text to compare our approach compared to the existing atlases in the discussion:

“Across 7 ages, the DevCCF contains 136 total samples (66 MRI, 70 LSFM), which is then doubled by flipping each sample for symmetry. Some developmental MRI atlases approach DevCCF sample sizes, but are restricted to MRI modality, approximately half DevCCF resolution with limited segmentations at each age (Table 1)^{34,57}. The 3D-reconstructed ADMBA meets DevCCF resolution and breadth of ages, but is based on a single 2D sectioned sample³³. In contrast, DevCCF contains average templates with sharper contrast that can represent overall morphology better than individual samples alone.”

Our goal is to create representative templates from both males and females in each age. Hence, we did not generate male or female templates separately. However, our averaged template can still be used to quantify potential sex differences and other changes by registering our common templates to individual samples. We added the following text in the result section.

“Averaging co-registered samples results in DevCCF templates with reduced individual noise and enhanced regional boundaries (Supplementary Fig. 1). Moreover, symmetrical templates can serve as initial space to assess potential laterality and sexual dimorphism of the brain by registering to individual samples.”

2. The authors used the DevCCF to map the migration of GABAergic cells (Figure 4), but it is unclear how this mapping reveals anything new on the GABAergic cell migration and the authors do not interpret their findings in light of the literature.

The main purpose of the figure is to demonstrate the utility of DevCCF to visualize and quantify fluorescently labeled cell types in developing mouse brains. We chose GABAergic neurons due to their significance in establishing excitation and inhibition balance to create mature neural circuits and their involvement in many brain disorders. Unlike relatively well-studied cortical interneurons, how GABAergic neurons emerge throughout the whole brain remains unclear. Moreover, prior studies largely rely on 2D histological sections, presenting limitation of cell type emergence in the whole 3D brains. Using GAD2-Cre;Ai14 data mapped onto the age matched DevCCF, we provide detailed description of how GABAergic neurons emerge from embryonic day 11.5 to 15.5. Furthermore, we interpreted our results in the context of prior studies. We added the following text in the results section “Charting early emergence of GABAergic neurons using DevCCF” and updated Figure 5 with more details.

“GABAergic cells are excitatory during development and become inhibitory in the mature brain, playing a key role in maintaining excitatory and inhibitory balance⁴⁴. While the developmental origin of cortical GABAergic interneurons is relatively well-studied^{45–47}, how GABAergic neurons emerge in the whole brain during early embryonic development remains under-studied.

... “We observed tdTomato labeled GABAergic neurons in samples from all three ages (Fig. 5a-c). We registered each sample to the aged-matched DevCCF template to create an averaged image per age (Fig. 5d-g).”

... “At E13.5, there is a rapid growth and expansion of GABAergic neurons in each cluster such as the hindbrain (h) including rhombomere 0 (r0) from the r1B and the subpallium septum (SeSPall) and striatum (Str) from the SPall (Fig. 5e, g). Moreover, GABAergic neurons newly appear in the cerebellar hemisphere (CbH) and midbrain tegmental area (MTg) (Fig. 5e, g). While p1 pretectal and p3 prethalamic areas contain high GABAergic neurons, prosomere 2 for the thalamus contains little to no GABAergic neurons (Fig. 5e, g). At E15.5, there are marked increases of GABAergic cells in all clusters as well as the newly emerging olfactory bulb (Fig. 5f, g). In addition to local expansion of GABAergic neurons, our mapping clearly visualizes chains of migrating cortical interneurons from the SPall to the superficial and deep cortical areas, reaching the half of the neocortex at E13.5 and fully reaching both the neocortex and the hippocampus at E15.5 (Fig. 5h), consistent with previous reports^{46,48}. Moreover, we found that migrating interneurons emerge in the intermediate and posterior neocortical areas at E13.5 and reached to the anterior area at E15.5 (Fig. 5h). Using DevCCF segmentations, quantitative analysis revealed the canonical spatiotemporal emergence of GABAergic neurons in developing brains (Fig. 5i).

Hence, we demonstrate that our DevCCF can serve as a standard spatial framework to visualize and quantify spatiotemporal trajectories of fluorescently labeled cell types in developing mouse brains.”

3. The authors show in Figure 5 a comparison between DevCCF and CCFv3: this is an important comparison but this aspect of the work is not fully explored in the manuscript. Here the authors could do a more ambitious comparison between the two atlases to clearly show where they agree or disagree, and present some evidence for example using the ISH data on the reasons from discrepancies between atlases.

The CCFv3 is based on adult cytoarchitecture, while the DevCCF is based on developmental gene expression. This different delineation criteria led to a few noticeable differences between the two labels. To describe this, we added a new supplementary Figure 6 and updated the following in Results section, ‘Integration of CCFv3 with P56 DevCCF for data analysis with a developmental framework’

“The CCFv3 thalamus encompasses the DevCCF thalamus (p2A; alar plate of p2) and prethalamus (PTh in prosomere 3; p3), while DevCCF pretectum (in prosomere 1; p1) maps to the CCFv3 midbrain (Fig. 6b, Supplementary Fig. 6a,b). Because p1 expresses early diencephalic marker Pax649, it belongs to the diencephalon (rather than the midbrain) in the DevCCF (Supplementary Figure 6c-e). Similarly, within DevCCF neuromeres (including p1, p2, and p3), we see separate segments of the substantia nigra due to early developmental neuromeric segmentations (Supplementary Fig. 6f,g). In contrast, the CCFv3 does not define developmental neuromeres, thereby segmenting the substantia nigra pars reticulata as one midbrain region (Supplementary Fig. 6g).”

Furthermore, we performed a systematic comparison to quantify how individual DevCCF labels overlap with CCFv3 labels and presented the data as Supplementary Table 4 to compare voxel-to-voxel correspondence between the two atlas labels.

“We quantified anatomical label correspondence between DevCCF and CCFv3 to facilitate further comparison of the two anatomical labels with different delineation criteria (Supplementary Table 4)”

A combination of this table and web visualization to overlay both atlases enables readers to gain a comprehensive understanding on how these two atlases compare to each other. We believe that both

atlases with different classification criteria are useful and together can be used to contextualize the developmental origins of adult structures. To highlight this point, we added the following text in the discussion.

“... This allows users to toggle between CCFv3 and DevCCF segmentations on our interactive web platform, providing information on how CCFv3 defined areas can be reclassified in DevCCF based on the developmental origin. We also show that transcriptional identity subclasses have concordance with anatomical boundaries in both atlases. For example, MERFISH derived transcriptional subclasses⁵⁰ of the DevCCF PPH align with two distinct parent regions in CCFv3: the cerebellum and part of the pons. While the cerebellum appears cytoarchitecturally distinct from the pons by adulthood, this corresponds with our knowledge about the development of the medial cerebellar vermis from the isthmic rhombomere (r0) and the lateral cerebellar hemisphere from rhombomere 142, and with our PPH growth curve confirming delayed growth as in previous literature³⁴. We argue that neither ontology is superior. Rather, an ontology should be selected based on user interest (e.g., adult cytoarchitecture vs developmental origin).”

4. The authors performed an interesting analysis and compared the DevCCF with the CCFve reference atlas, focusing on mapping cell type classification using a MERFISH dataset – this comparison did not directly address how cell type composition can be used to refine or adjust segmentation or definition of brain regions. Again, the analysis seems a bit superficial and lacks any reflection on findings or how to interpret them.

Thank you for providing this opportunity to expand our result and discussion. We added two examples of how MERFISH spatial transcriptome data can be interpreted with DevCCF and CCFv3. We found that a part of MERFISH labeled hypothalamic GABAergic neuronal cell types can be better aligned with DevCCF diencephalon labels and provide an opportunity to re-assign the zona incerta as diencephalic structure rather than being in the hypothalamic area. Moreover, DevCCF delineation helps to see the segregation of GABAergic and glutamatergic neurons in the hypothalamus. We updated Fig. 5 and added the following text in the result and discussion section.

In the results,

“... hypothalamic GABAergic cell types (12 HY GABA) are densely located in the reticular thalamus (RT), the zona incerta (ZI), and dorsal hypothalamic area while hypothalamic glutamatergic neurons (14 HY Glut) clustered in the ventral hypothalamus in the CCFv3 (Fig. 6f-g). In contrast, both the RT and the ZI are considered as the p3, a part of the diencephalon based on the developmental ontology, which is additionally supported by similar cell type distribution of GABAergic neurons (Fig. 6f-g). Moreover, DevCCF divides the hypothalamus into peduncular hypothalamus (PHy) and terminal hypothalamus (THy), which is populated by 12 HY GABA and 14 HY Glut, respectively. Similarly, the DevCCF considers the cerebellum as a part of the prepontine hindbrain (PPH) rather than a separate structure (as in CCFv3) because the cerebellum emerges in the rhombomere 0 and 1 as a part of the PPH during development (Fig 6c,e)⁵¹.

Hence, DevCCF offers new opportunities to re-interpret brain areas and emerging spatial genomic data in the context of developmental ontology.”

In the discussion we add,

“... We also show that transcriptional identity subclasses have concordance with anatomical boundaries in both atlases. For example, MERFISH derived transcriptional subclasses⁵⁰ of the DevCCF PPH align with two distinct parent regions in CCFv3: the cerebellum and part of the pons. While the cerebellum appears cytoarchitecturally distinct from the pons by adulthood, this corresponds with our knowledge about the development of the medial cerebellar vermis from the isthmic rhombomere (r0) and the lateral cerebellar hemisphere from rhombomere 142, and with our PPH growth curve confirming delayed growth as in previous literature³⁴. We argue that neither ontology is superior. Rather, an ontology should be selected based on user interest (e.g., adult cytoarchitecture vs developmental origin).”

“... Once integrated, these large multimodal datasets can be used with computational tools to unveil complex developmental mechanisms and drive automated segmentation⁸ at a scale not possible by human eye alone.”

5. To directly address the issue of how different biological signals can be used to define brain subregions, the authors should compare their DevCCF with the molecular atlas based on spatial transcriptomics (Ortiz et al, 2020), which defined cortical and subcortical regions based on the gene expression patterns in the adult mouse brain – it would be of great interest to compare the developmentally-defined regions in DevCCF in the adult (P56) with the regions defined by spatial transcriptomics to conclude how these two different definitions converge or diverge. The authors raised this important issue in their Discussion: to what extent different atlases agree on definitions and segmentation of subregions, and should take advantage of their DevCCF to provide some new insight into this question.

Thank you for the excellent suggestion to incorporate molecular atlas in our analysis. We were able to import it to the DevCCF template and cross compare with the DevCCF labels at P56 and the CCFv3 labels.

Segmentation for each atlas is based on different criteria, resulting in differences in borders of individual areas and ontological grouping of related areas (e.g., hypothalamic areas); 1) DevCCF (current manuscript) using genoarchitecture and prosomeric model, 2) CCFv3 using cytoarchitecture, and 3) Molecular atlas using genetic similarity across space. We found similarity and difference different criteria.

These discrepancies across different atlases highlight the importance of clearly indicating segmentation criteria for each atlas. Based on experimental questions, different atlases can be chosen. For instance, since the DevCCF segmentation is based on developmental origin of different brain areas, it can be advantageous to compare how molecular/cellular/anatomical features emerge across time from embryonic to adult brains with consistent ontological framework. In contrast, anatomical segmentations defined by molecular similarities can help to identify areas potentially with functional similarities.

We provide comparison of DevCCF and molecular atlas in supplementary figure 7 and added the following text in the discussion.

“For instance, the molecular atlas of the adult mouse brain utilized spatial transcriptomics to identify brain areas with similar molecular profiles and created anatomical labels with automated segmentations⁸. When integrated and compared to the DevCCF P56 template, we found both shared and diverging segmentations due to different delineation criteria (Supplementary Fig. 7). The DevCCF is an ideal atlas to map and integrate the growing quantity of developmental mouse brain spatial transcriptomics data^{11,57} and facilitate interoperability of different studies.”

REVIEWERS' COMMENTS

Reviewer #1 (Remarks to the Author):

The revision has improved. I would suggest the authors read more carefully about previous papers. IMHO, registering MRI data is a more or less solved problem in the field, thus registering the developmental mouse MRI data as shown in this manuscript has very limited technical novelty. It will be useful to cite previous work in an extended framework, not just mouse brain, but other mammalian systems as well (including human brains as suggested, including but not limited to the pointed literature), followed by highlighting the value of aligning the developmental series and explain why this research would be interesting and novel.

Reviewer #2 (Remarks to the Author):

The revised paper has improved considerably. It is appreciated that additional template files are made available and that new Figures have been added.

Hopefully the atlas will both be widely used and in due time also improved, corrected and updated. For this strict version control is needed with persistent identifiers. DOI and RRID should be provided before the publication. Rather than using FigShare, I recommended following the Data Repository Guidance for neuroscience provided by Nature:

<https://www.nature.com/sdata/policies/repositories#neurosci>

While important information about license and versioning is mentioned in the Data Availability statement and provided in Supp. Table 6, this information is not present in the web page from which the files are shared. On the contrary, a folder name states “DevCCF_MRI_sharable_v3.7_Public”, which is not related to the information in Supp Table 6. None of the files are labelled with version name, and it is very unlikely that users of the atlas will find, understand, and follow the versioning system given in Supp. Table 6. It is important that versioning information and instructions for citation are provided together with the data.

Concerning symmetric delineations, I maintain that I as a user would appreciate having an option for showing bilateral delineations, and a one-click option for toggling delineation visibility on and off. In this way it will be easier to observe and appreciate correspondences between the underlying anatomy and delineations.

The revised paper is clearly presented and well written, but the text has potential for further improvement.

Some minor comments.

Line 42-43, the list gives entities at different levels, in principle structure and function covers everything, connectivity, molecules and cells are part of the neural structure. Consider rephrasing.

Line 45, 47, introduce abbreviations 2D and 3D

Line 45-46, historical atlases typically were annotated diagrams based on serial 2D histologically stained sections, not “2D sections with histological staining”. Consider rephrasing

Line 47, interpreting anatomical regions in 3D brains. All brains are 3D by nature (actually histological sections also have a thickness), consider rephrasing to specify e.g. ... in brain image volumes, or 3D brain images / representations.

Line 50, “large scale data sets” is ambiguous -- large data or large amounts of data? Also consider adding that researchers produce increasing amounts of data from mouse models at different ages, to point to the need for atlases covering different stages than adult.

Line 90, use past tense (required)

Line 100, specify type of registration, linear or non-linear.

Line 108, specify “fixed”, formaldehyde-fixed?

Line 144, which anatomical landmarks were used. Can they be specified or listed? Are these the landmarks provided in Table 3?

Supplementary Figure 2 shows how the atlases were aligned to stereotaxic coordinates. It is good that the method used to define the tilt of the brain is provided, but as evident from the other figures

this orientation changes considerably across developmental ages. It is not clear how a user can make use of this information to conduct a stereotaxic experiment only using bregma for navigation. This tilt variation will influence the accuracy of stereotaxic surgery. This is indeed challenging to resolve, but the problem should at least be addressed in the Discussion.

Lines 235-237, how was this comparison done? Moreover, Supplementary File 4 is not review, not clear to me which file this is.

Line 334, while the atlas certainly is a valuable resource, I do not agree that it is an ideal atlas and suggest rephrasing. This is also emphasized by the authors in line 339.

Reviewer #3 (Remarks to the Author):

The authors have revised and considerably improved their manuscript in line with the reviewers' comments, and I think that the DevCCF can be a valuable resource for the field. I do not have any major questions or concerns.

REVIEWERS' COMMENTS

Reviewer #1 (Remarks to the Author):

The revision has improved. I would suggest the authors read more carefully about previous papers. IMHO, registering MRI data is a more or less solved problem in the field, thus registering the developmental mouse MRI data as shown in this manuscript has very limited technical novelty. It will be useful to cite previous work in an extended framework, not just mouse brain, but other mammalian systems as well (including human brains as suggested, including but not limited to the pointed literature), followed by highlighting the value of aligning the developmental series and explain why this research would be interesting and novel.

We appreciate reviewer's comment on the registration. We would like to emphasize the focus of the manuscript is the creation of developmental templates and associate 3D labels. With well-established image registration tools, multimodal data can be integrated in our new atlas framework. We added the following sentence in our discussion with added citations.

"MRI has been used to examine macroscopic changes in brain morphology, connectivity, and function **with well-established registration methods in humans and other species**^{29,63-71}"

We indeed emphasize the value of developmental atlases and the need for similar works in other species, such as following texts in the discussion.

"Importantly, grounded in the prosomeric model of vertebrate development, the DevCCF can serve as the foundation for constructing developmental atlases for other mammalian species (e.g., macaque, human)⁸³⁻⁸⁵"

Reviewer #2 (Remarks to the Author):

The revised paper has improved considerably. It is appreciated that additional template files are made available and that new Figures have been added. Hopefully the atlas will both be widely used and in due time also improved, corrected and updated.

- For this strict version control is needed with persistent identifiers. DOI and RRID should be provided before the publication. Rather than using FigShare, I recommended following the Data Repository Guidance for neuroscience provided by Nature: <https://www.nature.com/sdata/policies/repositories#neurosci>

Thank you for this suggestion. At this point in time, the DevCCF is shared via FigShare with an associated DOI (<https://doi.org/10.6084/m9.figshare.26377171>). We have also included the RRID (SCR_025544) in this manuscript. In order to avoid data duplication and confusion we have decided not to duplicate the downloadable DevCCF to second location. FigShare is a generalist repository for open and FAIR data recommended on the same Nature webpage shared above. It includes repository metadata, DOI generation, citation information, version tracking, and licensing. From here, the DevCCF can later be linked to repositories such as the Neuroimaging Informatics Tools and Resources Collaboratory (NITRC) and EBRAINS to be sure it is accessible in neuroscience specific domains.

- While important information about license and versioning is mentioned in the Data Availability statement and provided in Supp. Table 6, this information is not present in the web page from which the files are shared. On the contrary, a folder name states “DevCCF_MRI_sharable_v3.7_Public”, which is not related to the information in Supp Table 6. None of the files are labelled with version name, and it is very unlikely that users of the atlas will find, understand, and follow the versioning system given in Supp. Table 6. It is important that versioning information and instructions for citation are provided together with the data.

Thank you for your efforts to ensure the DevCCF is FAIR and accessible. The web page link you are referring to (“Beta Release Download”) goes to the “DevCCF_MRI_sharable_v3.7_Public” folder, which is a beta version from the initial submission (DevCCF beta 3.7). This is the dataset from which your notes on initial revision were based. DevCCFv1 is available for download via FigShare (<https://doi.org/10.6084/m9.figshare.26377171>). This dataset includes a DevCCFv1_ReadMe.md file that details DevCCF download file descriptions, abbreviations, elements list, licensing info, citation instructions, support, and version release notes. The parent folder is called DevCCFv1. We plan to update the downloadable link in our website with the latest version (currently v1).

- Concerning symmetric delineations, I maintain that I as a user would appreciate having an option for showing bilateral delineations, and a one-click option for toggling delineation visibility on and off. In this way it will be easier to observe

and appreciate correspondences between the underlying anatomy and delineations.

On the DevCCF interactive web visualization, users may click the eye icon to toggle any layer visibility on and off, including annotations layers. The eye icon can be seen in Fig. 7a. Half brain segmentations enable users to observe and appreciate correspondences between the underlying anatomy and delineations by toggling, or they can simultaneously view delineations on one hemisphere and compare to the underlying symmetric anatomy across the midline. As requested in the previous revision, we have generated bilateral delineations which are available in the FigShare repository. These can be downloaded and inspected in the user image viewer of choice.

The revised paper is clearly presented and well written, but the text has potential for further improvement. Some minor comments.

- Line 42-43, the list gives entities at different levels, in principle structure and function covers everything, connectivity, molecules and cells are part of the neural structure. Consider rephrasing.

Thank you for identifying this. We have edited the statement to read, "Brain atlases provide a standard anatomical context to interpret brain structure and function, including neuronal connectivity, molecular signatures, and cell type specific transcriptome data."

- Line 45, 47, introduce abbreviations 2D and 3D

We have introduced 2-dimensional (2D) and 3-dimensional (3D) acronym terms.

- Line 45-46, historical atlases typically were annotated diagrams based on serial 2D histologically stained sections, not "2D sections with histological staining". Consider rephrasing

This statement has been rephrased as suggested

- Line 47, interpreting anatomical regions in 3D brains. All brains are 3D by nature (actually histological sections also have a thickness), consider rephrasing to specify e.g. ... in brain image volumes, or 3D brain images / representations.

We have rephrased to read, "interpreting anatomical regions in 3-dimensional (3D) brain imaging data"

- Line 50, "large scale data sets" is ambiguous -- large data or large amounts of data? Also consider adding that researchers produce increasing amounts of data from mouse models at different ages, to point to the need for atlases covering different stages than adult.

Modified to read, "allow researchers to rapidly produce increasing amounts of 3D data from mouse models at various ages."

- Line 90, use past tense (required)

This sentence has been modified past tense.

- Line 100, specify type of registration, linear or non-linear.

We specify that "LSFM templates were non-linearly registered to the MRI templates".

- Line 108, specify "fixed", formaldehyde-fixed?

Added detail to describe "paraformaldehyde-fixed ex-vivo in-skull samples"

- Line 144, which anatomical landmarks were used. Can they be specified or listed? Are these the landmarks provided in Table 3?

We altered this sentence to read, "We used landmarks (Table 3) that are visible in DevCCF templates (Fig. 3, Supplementary Fig. 4), as well as aligned 3D imaging and side-by-side reference materials as evidence to draw 3D anatomical segmentations.", specifying that landmark examples are listed in Table 3 and viewed in Fig. 3 and Supplementary Fig 4.

- Supplementary Figure 2 shows how the atlases were aligned to stereotaxic coordinates. It is good that the method used to define the tilt of the brain is provided, but as evident from the other figures this orientation changes considerably across developmental ages. It is not clear how a user can make use of this information to conduct a stereotaxic experiment only using bregma for navigation. This tilt variation will influence the accuracy of stereotaxic surgery. This is indeed challenging to resolve, but the problem should at least be addressed in the Discussion.

Modified Results section 3D Multimodal Developmental Mouse Brain Templates to: "Moreover, we established the postnatal templates with stereotaxic coordinates by aligning estimated bregma and anterior commissure locations from MRI templates (Supplementary Fig. 2a). 3D stereotaxic coordinate grid images (Supplementary Fig. 2b) were generated for each postnatal age that can be overlaid on templates to facilitate in vivo injections or recording experiments in the future"

in Discussion: "Stereotaxically aligned postnatal templates with labelled coordinate grid overlays provide spatial coordinates to guide surgical procedures (e.g., stereotaxic brain injection). ... As postnatal templates do not include skull data, stereotaxic coordinates are defined by anterior commissure and estimated bregma, reducing the accuracy

anterior-posterior alignment during in vivo stereotaxic procedures. Adding skull data to postnatal templates will enable lambda and bregma defined stereotaxic coordinates, improving alignment for such procedures. "

- Lines 235-237, how was this comparison done? Moreover, Supplementary File 4 is not review, not clear to me which file this is.

This section was edited to note "overlapping voxel-to-voxel anatomical label correspondence". The corresponding first paragraph of section Methods: Integrating CCFv3 with Spatial Transcriptomics and molecular atlas to DevCCF, and was reworded to mirror the language of the results paragraph.

- Line 334, while the atlas certainly is a valuable resource, I do not agree that it is an ideal atlas and suggest rephrasing. This is also emphasized by the authors in line 339.

This has been edited to read, "The DevCCF can serve as a resource to map and integrate the growing quantity of developmental mouse brain spatial transcriptomics data^{11,57} and facilitate interoperability between different studies"

Reviewer #3 (Remarks to the Author):

The authors have revised and considerably improved their manuscript in line with the reviewers' comments, and I think that the DevCCF can be a valuable resource for the field. I do not have any major questions or concerns.

Thank you for your thoughtful comments that have helped improve this manuscript and the DevCCF.